# L2E: Learning to Exploit Your Opponent

## Abstract

Opponent modeling is essential to exploit sub-optimal opponents in strategic interactions. One key challenge facing opponent modeling is how to fast adapt to opponents with diverse styles of strategies. Most previous works focus on building *explicit* models to predict the opponents' styles or strategies directly. However, these methods require a large amount of data to train the model and lack the adaptability to new opponents of unknown styles. In this work, we propose a novel Learning to Exploit (L2E) framework for *implicit* opponent modeling. L2E acquires the ability to exploit opponents by a few interactions with different opponents during training so that it can adapt to new opponents with unknown styles during testing quickly. We propose a novel Opponent Strategy Generation (OSG) algorithm that produces effective opponents for training automatically. By learning to exploit the challenging opponents generated by OSG through adversarial training, L2E gradually eliminates its own strategy's weaknesses. Moreover, the generalization ability of L2E is significantly improved by training with diverse opponents, which are produced by OSG through diversity-regularized policy optimization. We evaluate the L2E framework on two poker games and one grid soccer game, which are the commonly used benchmark for opponent modeling. Comprehensive experimental results indicate that L2E quickly adapts to diverse styles of unknown opponents.

## 1 Introduction

One core research topic in modern artificial intelligence is creating agents that can interact effectively with their opponents in different scenarios. To achieve this goal, the agents should have the ability to reason about their opponents' behaviors, goals, and beliefs. Opponent modeling, which constructs the opponents' models to reason about them, has been extensively studied in past decades (Albrecht & Stone, 2018). In general, an opponent model is a function that takes some interaction history as its input and predicts some property of interest of the opponent. Specifically, the interaction history may contain the past actions that the opponent took in various situations, and the properties of interest could be the actions that the opponent may take in the future, the style of the opponent (*e.g.*, "defensive", "aggressive"), or its current goals. The resulting opponent model can inform the agent's decision-making by incorporating the model's predictions in its planning procedure to optimize its interactions with the opponent. Opponent modeling has already been used in many practical applications, such as dialogue systems (Grosz & Sidner, 1986), intelligent tutor systems (McCalla et al., 2000), and security systems (Jarvis et al., 2005).

Many opponent modeling algorithms vary greatly in their underlying assumptions and methodology. For example, policy reconstruction based methods (Powers & Shoham, 2005; Banerjee & Sen, 2007) explicitly fit an opponent model to reflect the opponent's observed behaviors. Type reasoning based methods (Dekel et al., 2004; Nachbar, 2005) reuse pre-learned models of several known opponents by finding the one which most resembles the behavior of the current opponent. Classification based methods (Huynh et al., 2006; Sukthankar & Sycara, 2007) build models that predict the play style of the opponent, and employ the counter-strategy, which is effective against that particular style. Some recent works combine opponent modeling with deep learning methods or reinforcement learning methods and propose many related algorithms (He et al., 2016; Foerster et al., 2018; Wen et al., 2018). Although these algorithms have achieved some success, they also have some obvious disadvantages. First, constructing accurate opponent models requires a lot of data, which is problematic since the agent does not have the time or opportunity to collect enough data about its opponent in

most applications. Second, most of these algorithms perform well only when the opponents during testing are similar to the ones used for training, and it is difficult for them to adapt to opponents with new styles quickly. More related works on opponent modeling are in Appendix A.1.

To overcome these shortcomings, we propose a novel Learning to Exploit (L2E) framework in this work for *implicit* opponent modeling, which has two desirable advantages. First, L2E does not build an explicit model for the opponent, so it does not require a large amount of interactive data and eliminates the modeling errors simultaneously. Second, L2E can quickly adapt to new opponents with unknown styles, with only a few interactions with them. The key idea underlying L2E to train a *base policy* against various styles of opponents by using only a few interactions between them during training, such that it acquires the ability to exploit different opponents quickly. After training, the base policy can quickly adapt to new opponents using only a few interactions during testing. In effect, our L2E framework optimizes for a base policy that is easy and fast to adapt. It can be seen as a particular case of learning to learn, *i.e.*, meta-learning (Finn et al., 2017). The meta-learning algorithm (*c.f.*, Appendix A.2 for details), such as MAML (Finn et al., 2017), is initially designed for single-agent environments. It requires manual design of training tasks, and the final performance largely depends on the user-specified training task distribution. The L2E framework is designed explicitly for the multi-agent competitive environments, which generates effective training tasks (opponents) automatically (*c.f.*, Appendix A.3 for details). Some recent works have also initially used meta-learning for opponent modeling. Unlike these works, which either use meta-learning to predict the opponent's behaviors (Rabinowitz et al., 2018) or to handle the non-stationarity problem in multi-agent reinforcement learning (Al-Shedivat et al., 2018), we focus on how to improve the agent's ability to adapt to unknown opponents quickly.

In our L2E framework, the base policy is explicitly trained such that a few interactions with a new opponent will produce an *opponent-specific* policy to effectively exploit this opponent, *i.e.*, the base policy has strong adaptability that is broadly adaptive to many opponents. In specific, if a deep neural network models the base policy, then the opponent-specific policy can be obtained by fine-tuning the parameters of the base policy's network using the new interactive data with the opponent. A critical step in L2E is how to generate effective opponents to train the base policy. The ideal training opponents should satisfy the following two desiderata. 1) The opponents need to be challenging enough (*i.e.*, hard to exploit). By learning to exploit these challenging opponents, the base policy eliminates its weakness and learns a more robust strategy. 2) The opponents need to have enough diversity. The more diverse the opponents during training, the stronger the base policy's generalization ability is, and the more adaptable the base policy to the new opponents.

To this end, we propose a novel opponent strategy generation (OSG) algorithm, which can produce challenging and diverse opponents automatically. We use the idea of adversarial training to generate challenging opponents. Some previous works have also been proposed to obtain more robust policies through adversarial training and showed that it improves the generalization (Pinto et al., 2017; Pattanaik et al., 2018). From the perspective of the base policy, giving an opponent, the base policy first adjusts itself to obtain an adapted policy, the base policy is then optimized to *maximize* the rewards that the adapted policy gets when facing the opponent. The challenging opponents are then adversarially generated by *minimizing* the base policy's adaptability by automatically generating difficult to exploit opponents. These hard-to-exploit opponents are trained such that even if the base policy adapts to them, the adapted base policy cannot take advantage of them. Besides, our OSG algorithm can further produce diverse training opponents with a novel diversity-regularized policy optimization procedure. In specific, we use the Maximum Mean Discrepancy (MMD) metric (Gretton et al., 2007) to evaluate the differences between policies. The MMD metric is then incorporated as a regularization term into the policy optimization process to obtain a diverse set of opponent policies. By training with these challenging and diverse training opponents, the robustness and generalization ability of our L2E framework can be significantly improved. To summarize, the main contributions of this work are listed bellow in four-fold:

- We propose a novel learning to exploit (L2E) framework to exploit sub-optimal opponents without building explicit models for it. L2E can quickly adapt to a new opponent with unknown style using only a few interactions.

- We propose to use an adversarial training procedure to generate challenging opponents automatically. These hard to exploit opponents help L2E eliminate its weakness and improve its robustness effectively.

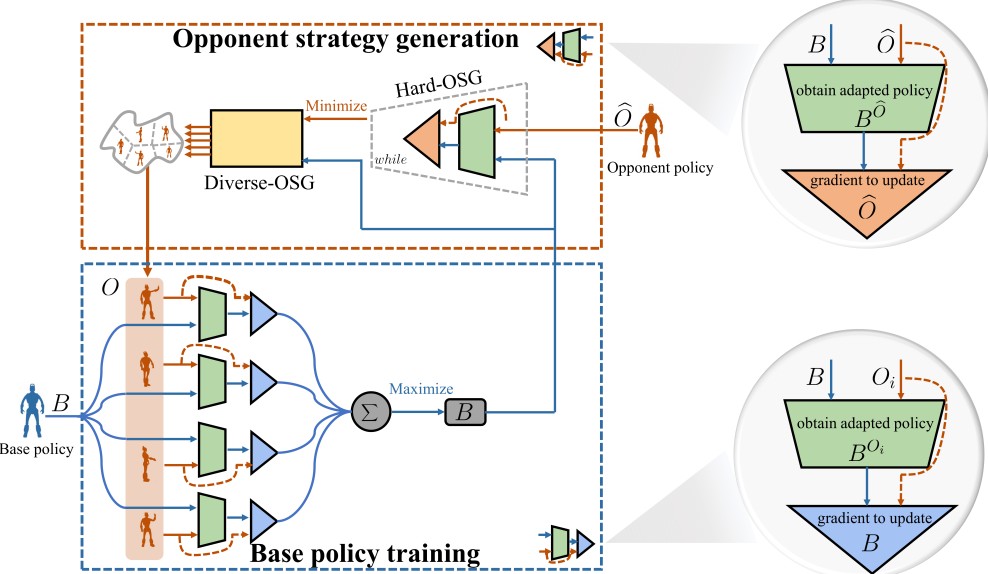

Figure 1: The overview of our proposed L2E framework. The entire training process is based on the idea of adversarial learning (Alg. 1). The base policy training part maximizes the base policy's adaptability by continually interacting with opponents of different strengths and styles (Section 2.1). The opponent strategy generation part first generates hard-to-exploit opponents for the current base policy (Hard-OSG, see Section 2.2.1), then generates diverse opponent policies to improve the generalization ability of the base policy (Diverse-OSG, see Section 2.2.2). The resulting base policy can fast adapt to completely new opponents with a few interactions.

- We further propose a diversity-regularized policy optimization procedure to generate diverse opponents automatically. The generalization ability of L2E is improved significantly by training with these diverse opponents.

- We conduct detailed experiments to evaluate the L2E framework in three different environments. The experimental results demonstrate that the base policy trained with L2E quickly exploits a wide range of opponents compared to other algorithms.

## 2 METHOD

In this paper, we propose a novel L2E framework to endow the agents to adapt to diverse opponents quickly. As shown in Fig. 1, L2E mainly consists of two modules, *i.e.*, the base policy training part, and the opponent strategy generation part. In the base policy training part, our goal is to find a base policy that, given the unknown opponent, can fast adapt to it by using only a few interactions. To this end, the base policy is trained to be able to adapt to many opponents. In specific, giving an opponent $O$, the base policy $B$ first adjusts itself to obtain an adapted policy $B'$ by using a little interaction data between $O$ and $B$, the base policy is then optimized to maximize the rewards that $B'$ gets when facing $O$. In other words, the base policy has learned how to adapt to its opponents and exploit them quickly.

The opponent strategy generation provides the base policy training part with challenging and diverse training opponents automatically. First, our proposed opponent strategy generation (OSG) algorithm can produce difficult to exploit opponents. In specific, the base policy $B$ first adjusts itself to obtain an adapted policy $B'$ by using a little interaction data between $O$ and $B$, the opponent $O$ is then optimized to *minimize* the rewards that $B'$ gets when facing $O$. The resulting opponent $O$ is hard to exploit since even if the base policy $B$ adapts to $O$, the adapted policy $B'$ can not take advantage of $O$. By training with these hard to exploit opponents, the base policy can eliminate its weakness and improve its robustness effectively. Second, our OSG algorithm can further produce diverse training opponents with a novel diversity-regularized policy optimization procedure. More specifically, we

first formalize the difference between opponent policies as the difference between the distribution of trajectories induced by each policy. The difference between distributions can be evaluated by the Maximum Mean Discrepancy (MMD) metric (Gretton et al., 2007). Then, MMD is integrated as a regularization term in the policy optimization process to identify various opponent policies. By training with these diverse opponents, the base policy can improve its generalization ability significantly. Next, we introduce these two modules in detail.

## 2.1 BASE POLICY TRAINING

Our goal is to find a base policy $B$ that can fast adapt to an unknown opponent $O$ by updating the parameters of $B$ using only a few interactions between $B$ and $O$. The key idea is to train the base policy $B$ against many opponents to maximize its payoffs by using only a small amount of interactive data during training, such that it acquires the ability to exploit different opponents quickly. In effect, our L2E framework treats each opponent as a training example. After training, the resulting base policy $B$ can quickly adapt to new and unknown opponents using only a few interactions. Without loss of generality, the base policy $B$ is modeled by a deep neural network in this work, *i.e.*, a parameterized function $\pi_\theta$ with parameters $\theta$. Similarly, the opponent $O$ for training is also a deep neural network $\pi_\phi$ with parameters $\phi$. We model the base policy as playing against an opponent in a two-player Markov game (Shapley, 1953). This Markov game $M = (S, (A_B, A_O), T, (R_B, R_O))$ consists of the state space $S$, the action space $A_B$ and $A_O$, and a state transition function $T : S \times A_B \times A_O \to \Delta(S)$ where $\Delta(S)$ is a probability distribution on $S$. The reward function $R_i : S \times A_B \times A_O \times S \to \mathbb{R}$ for each player $i \in \{B, O\}$ depends on the current state, the next state and both players' actions. Given a training opponent $O$ whose policy is known and fixed, this two-player Markov game $M$ reduces to a single-player Markov Decision Process (MDP), *i.e.*, $M_B^O = (S, A_B, T_B^O, R_B^O)$. The state and action space of $M_B^O$ are the same as in $M$. The transition and reward functions have the opponent policy embedded:

$$T_B^O(s, a_B) = T(s, a_B, a_O), \quad R_B^O(s, a_B, s') = R_B(s, a_B, a_O, s'),$$

where the opponent's action is sampled from its policy $a_O \sim \pi_\phi(\cdot \mid s)$. Throughout the paper, $M_X^Y$ represents a single-player MDP, which is reduced from a two-player Markov game (*i.e.*, player $X$ and player $Y$). In this MDP, the player $Y$ is fixed and can be regarded as part of the environment.

Suppose a set of training opponents $\{O_i\}_{i=1}^N$ is given. For each training opponent $O_i$, an MDP $M_B^{O_i}$ can be constructed as described above. The base policy $B$, *i.e.*, $\pi_\theta$ is allowed to query a limited number of sample trajectories $\tau$ to adapt to $O_i$. In our method, the adapted parameters $\theta^{O_i}$ of the base policy are computed using one or more gradient descent updates with the sample trajectories $\tau$. For example, when using one gradient update:

$$\theta^{O_i} = \theta - \alpha \nabla_\theta \mathcal{L}_B^{O_i}(\pi_\theta), \tag{1}$$

$$\mathcal{L}_B^{O_i}(\pi_\theta) = -\mathbb{E}_{\tau \sim M_B^{O_i}}\left[\sum_t \gamma^t R_B^{O_i}(s^{(t)}, a_B^{(t)}, s^{(t+1)})\right]. \tag{2}$$

$\tau \sim M_B^{O_i}$ represents that the trajectory $\tau = \{s^{(1)}, a_B^{(1)}, s^{(2)}, \ldots, s^{(t)}, a_B^{(t)}, s^{(t+1)}, \ldots\}$ is sampled from the MDP $M_B^{O_i}$, where $s^{(t+1)} \sim T_B^{O_i}(s^{(t)}, a_B^{(t)})$ and $a_B^{(t)} \sim \pi_\theta(\cdot \mid s^{(t)})$.

We use $B^{O_i}$ to denote the updated base policy, *i.e.*, $\pi_{\theta^{O_i}}$. $B^{O_i}$ can be seen as an *opponent-specific* policy, which is updated from the base policy through fast adaptation. Our goal is to find a generalizable base policy whose opponent-specific policy $B^{O_i}$ can exploit its opponent $O_i$ as much as possible. To this end, we optimize the parameters $\theta$ of the base policy to maximize the rewards that $B^{O_i}$ gets when interacting with $O_i$. More concretely, the *learning to exploit* objective function is defined as follows:

$$\min_\theta \sum_{i=1}^N \mathcal{L}_{B^{O_i}}^{O_i}(\pi_{\theta^{O_i}}) = \min_\theta \sum_{i=1}^N \mathcal{L}_{B^{O_i}}^{O_i}(\pi_{\theta - \alpha \nabla_\theta \mathcal{L}_B^{O_i}(\pi_\theta)}). \tag{3}$$

It is worth noting that the optimization is performed over the base policy's parameters $\theta$, whereas the objective is computed using the adapted based policy's parameters $\theta^{O_i}$. The parameters $\theta$ of the base policy are updated as follows:

$$\theta = \theta - \beta \nabla_\theta \sum_{i=1}^N \mathcal{L}_{B^{O_i}}^{O_i}(\pi_{\theta^{O_i}}). \tag{4}$$

In effect, our L2E framework aims to find a base policy that can significantly exploit the opponent with only a few interactions with it (*i.e.*, with a few gradient steps). The resulting base policy has learned how to adapt to different opponents and exploit them quickly. An overall description of the base policy training procedure is shown in Alg. 1. The algorithm consists of three main steps. First, generating hard to exploit opponents through the Hard-OSG module. Second, generating diverse opponent policies through the Diverse-OSG module. Third, training the base policy with these opponents to obtain fast adaptability.

## 2.2 AUTOMATIC OPPONENT GENERATION

Previously, we assumed that the set of opponents had been given. How to automatically generate effective opponents for training is the key to the success of our L2E framework. The training opponents should be challenging enough (*i.e.*, hard to exploit). By learning to exploit these hard-to-exploit opponents, the base policy $B$ can eliminate its weakness and become more robust. Besides, they should be sufficiently diverse. The more diverse they are, the stronger the generalization ability of the resulting base policy. We propose a novel opponent strategy generation (OSG) algorithm to achieve these goals.

### 2.2.1 HARD-TO-EXPLOIT OPPONENTS GENERATION

We use the idea of adversarial learning to generate challenging training opponents for the base policy $B$. From the perspective of the base policy $B$, giving an opponent $O$, $B$ first adjusts itself to obtain an adapted policy, *i.e.*, the opponent-specific policy $B^O$, the base policy is then optimized to *maximize* the rewards that $B^O$ gets when interacting with $O$. Contrary to the base policy's goal, we want to find a hard-to-exploit opponent $\widehat{O}$ for the current base policy $B$, such that even if $B$ adapts to $\widehat{O}$, the adapted policy $B^{\widehat{O}}$ cannot take advantage of $\widehat{O}$. In other words, the hard-to-exploit opponent $\widehat{O}$ is trained to *minimize* the rewards that $B^{\widehat{O}}$ gets when interacting with $\widehat{O}$. The base policy attempts to increase its adaptability by learning to exploit different opponents, while the hard-to-exploit opponent adversarially tries to minimize the base policy's adaptability, *i.e.*, maximize its *counter-adaptability*.

More concretely, the hard-to-exploit opponent $\widehat{O}$ is also a deep neural network $\pi_{\widehat{\phi}}$ with randomly initialized parameters $\widehat{\phi}$. At each training iteration, an MDP $M_B^{\widehat{O}}$ can be constructed. The base policy $B$ first query a limited number of trajectories to adapt to $\widehat{O}$. The parameters $\theta^{\widehat{O}}$ of the adapted policy $B^{\widehat{O}}$ are computed using one gradient descent update,

$$\theta^{\widehat{O}} = \theta - \alpha \nabla_\theta \mathcal{L}_B^{\widehat{O}}(\pi_\theta). \tag{5}$$

$$\mathcal{L}_B^{\widehat{O}}(\pi_\theta) = -\mathbb{E}_{\tau \sim M_B^{\widehat{O}}}[\sum_t \gamma^t R_B^{\widehat{O}}(s^{(t)}, a_B^{(t)}, s^{(t+1)})]. \tag{6}$$

The parameters $\widehat{\phi}$ of $\widehat{O}$ is optimized to minimize the rewards that $B^{\widehat{O}}$ gets when interacting with $\widehat{O}$. This is equivalent to maximize the rewards that $\widehat{O}$ gets since we only consider the competitive setting in this work. More concretely, the parameters $\widehat{\phi}$ are updated as follows:

$$\widehat{\phi} = \widehat{\phi} - \alpha \nabla_{\widehat{\phi}} \mathcal{L}_{\widehat{O}}^{B^{\widehat{O}}}(\pi_{\widehat{\phi}}) \tag{7}$$

$$\mathcal{L}_{\widehat{O}}^{B^{\widehat{O}}}(\pi_{\widehat{\phi}}) = -\mathbb{E}_{\tau' \sim M_{\widehat{O}}^{B^{\widehat{O}}}}[\sum_t \gamma^t R_{\widehat{O}}^{B^{\widehat{O}}}(s^{(t)}, a_{\widehat{O}}^{(t)}, s^{(t+1)})]. \tag{8}$$

After several rounds of iteration, we can obtain a hard-to-exploit opponent $\pi_{\widehat{\phi}}$ for the current base policy $B$. An overall description of this procedure is shown in Alg. 2.

### 2.2.2 DIVERSE OPPONENTS GENERATION

Training an effective base policy requires not only the hard-to-exploit opponents, but also diverse opponents of different styles. The more diverse the opponents used for training, the stronger the generalization ability of the resulting base policy. From a human player's perspective, the opponent style is usually defined as different types, such as aggressive, defensive, elusive, *etc*. The most significant difference between opponents with different styles lies in the actions taken in the same state.

Take poker as an example; different opponents' styles tend to take different actions when holding the same hand. Based on the above analysis, we formalize the difference between opponent policies as the difference between the distribution of trajectories induced by each policy when interacting with the base policy. We argue that differences in trajectories better capture the differences between different opponent policies.

Formally, given a base policy $B$, *i.e.*, $\pi_\theta$ and an opponent policy $O_i$, *i.e.*, $\pi_{\phi_i}$, our diversity-regularized policy optimization algorithm is to generate a new opponent $O_j$, *i.e.*, $\pi_{\phi_j}$ whose style is different from $O_i$. We first construct two MDPs, *i.e.*, $M_{O_i}^B$ and $M_{O_j}^B$, and then sample two sets of trajectories, *i.e.*, $\mathrm{T}_i = \{\tau \sim M_{O_i}^B\}$ and $\mathrm{T}_j = \{\tau \sim M_{O_j}^B\}$ from this two MDPs. The stochasticity in the MDP and the policy will induce a distribution over trajectories. We use the Maximum Mean Discrepancy (MMD) (Gretton et al., 2007) metric (*c.f.* Appendix C for details) to measure the differences between $\mathrm{T}_i$ and $\mathrm{T}_j$:

$$\mathrm{MMD}^2(\mathrm{T}_i, \mathrm{T}_j) = \mathbb{E}_{\tau,\tau' \sim M_{O_i}^B} k(\tau, \tau') - 2\mathbb{E}_{\tau \sim M_{O_i}^B, \tau' \sim M_{O_j}^B} k(\tau, \tau') + \mathbb{E}_{\tau,\tau' \sim M_{O_j}^B} k(\tau, \tau'). \quad (9)$$

$k$ is the Gaussian radial basis function kernel defined over a pair of trajectories:

$$k(\tau, \tau') = \exp(-\frac{\|g(\tau) - g(\tau')\|^2}{2}), \quad (10)$$

where $g$ stacks the states and actions of a trajectory into a vector. For trajectories with different length, we clip the long trajectory to the same length as the short one. There overall objective function of our proposed diversity-regularized policy optimization algorithm is as follows:

$$\mathcal{L}^{\phi_i}(\phi_j) = -\mathbb{E}_{\tau \sim M_{O_j}^B}[\sum_t \gamma^t R_{O_j}^B(s^{(t)}, a_{O_j}^{(t)}, s^{(t+1)})] - \alpha_{mmd} \mathrm{MMD}^2(\mathrm{T}_i, \mathrm{T}_j). \quad (11)$$

The first term is to maximize the rewards that $O_j$ gets when interacting with the base policy $B$. The second term measures the difference between $O_j$ and the existing opponent $O_i$. By this diversity-regularized policy optimization, the resulting opponent $O_j$ is not only useful in performance but also diverse relative to the existing policy.

We can iteratively apply the above algorithm to find a set of $N$ distinct and diverse opponents. In specific, subsequent opponents are learned by encouraging diversity with respect to previously generated opponent set $S$. The distance between an opponent $O_m$ and an opponent set $S$ is defined by the distance between $O_m$ and $O_n$, where $O_n \in S$ is the most similar policy to $O_m$. Suppose we have obtained a set of opponents $S = \{O_m\}_{m=1}^M, M < N$. The $M + 1$-th opponent, *i.e.*, $\pi_{\phi_{M+1}}$ can be obtained by optimizing:

$$\mathcal{L}^S(\phi_{M+1}) = -\mathbb{E}_{\tau \sim M_{O_{M+1}}^B}[\sum_t \gamma^t R_{O_{M+1}}^B(s^{(t)}, a_{O_{M+1}}^{(t)}, s^{(t+1)})] - \min_{O_i \in S} \mathrm{MMD}^2(\mathrm{T}_i, \mathrm{T}_{M+1}). \quad (12)$$

By doing so, the resulting $M + 1$-th opponent remains diverse relative to the opponent set $S$. An overall description of this procedure is shown in Alg. 3.

## 3 EXPERIMENTS

In this section, we conduct extensive experiments to evaluate the proposed L2E framework. We evaluate algorithm performance on the Leduc poker, the BigLeduc poker and a Grid Soccer environment, which are the commonly used benchmark for opponent modeling (Lanctot et al., 2017; Steinberger, 2019; He et al., 2016). We first verify the trained base policy using our L2E framework can fast exploit a wide range of opponents with only a few gradient updates. Then, we compare with other baseline methods to show the superiority of our L2E framework. Finally, we conduct a series of ablation experiments to demonstrate each part of our L2E framework's effectiveness.

### 3.1 RAPID ADAPTABILITY

In this section, we verify the trained base policy's ability to quickly adapt to different opponents in the Leduc poker environment (*c.f.* Appendix D for details). We provide four opponents with different styles and strengths. 1) The random opponent randomly takes actions whose strategy is

relatively weak but hard to exploit since it does not have an evident decision-making style. 2) The call opponent always takes call actions and has a fixed decision-making style that is easy to exploit. 3) The rocks opponent takes actions based on its hand-strength whose strategy is relatively strong. 4) The oracle opponent is a cheating, and the strongest player who can see the other players' hands and make decisions based on this perfect information. As shown in Fig. 2, the base policy achieves a

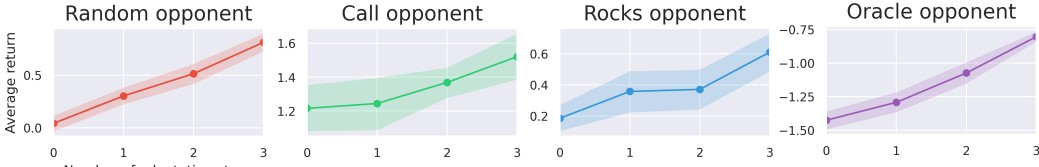

Figure 2: The trained base policy using our L2E framework can quickly adapt to different opponents of different styles and strengths in the Leduc poker environment.

rapid increase in its average returns with a few gradient updates against all four opponent strategies. For the call opponent, which has a clear and monotonous style, the base policy can significantly exploit it. Against the random opponent with no clear style, the base policy can also exploit it quickly. When facing the strong rocks opponent or even the strongest oracle opponent, the base policy can quickly improve its average returns. One significant advantage of the proposed L2E framework is that the same base policy can exploit a wide range of opponents with different styles, demonstrating its strong generalization ability.

## 3.2 COMPARISONS WITH OTHER BASELINE METHODS

|  | Random | Call | Rocks | Nash | Oracle |
|---|---|---|---|---|---|
| L2E | 0.42±0.32 | **1.34±0.14** | **0.38±0.17** | **-0.03±0.14** | **-1.15±0.27** |
| MAML | **1.27±0.17** | -0.23±0.22 | -1.42±0.07 | -0.77±0.23 | -2.93±0.17 |
| Random | -0.02±3.77 | -0.02±3.31 | -0.68±3.75 | -0.74±4.26 | -1.90±4.78 |
| TRPO | 0.07±0.08 | -0.22±0.09 | -0.77±0.12 | -0.42±0.07 | -1.96±0.46 |
| TRPO (pretrained) | 0.15±0.17 | -0.05±0.14 | -0.70±0.27 | -0.61±0.32 | -1.32±0.27 |
| EOM (Explicit Opponent Modeling) | 0.30±0.15 | -0.01±0.05 | -0.13±0.20 | -0.36±0.11 | -1.82±0.28 |

Table 1: The average return of each method when performing rapid adaptation against different opponents in the Leduc Poker environment. The adaptation process is restricted to a three-step gradient update.

As discussed in Section 1, most previous opponent modeling methods require constructing explicit opponent models from a large amount of data before learning to adapt to new opponents. To the best of our knowledge, our L2E framework is the first attempt to use meta-learning to learn to exploit opponents without building explicit opponent models. To demonstrate the effectiveness of the L2E framework, we design several competitive baseline methods. As with the previous experiments, we also use three gradient updates when adapting to a new opponent. 1) MAML. The seminal meta-learning algorithm MAML (Finn et al., 2017) is designed for single-agent environments. We have redesigned and reimplemented the MAML algorithm for the two-player competitive environments. The MAML baseline trains a base policy by continually sampling the opponent's strategies, either manually specified or randomly generated. 2) TRPO. The TRPO baseline does not perform pre-training and uses the TRPO algorithm (Schulman et al., 2015) to updated its parameters via three-step gradient updates to adapt to different opponents. 3) Random. The Random baseline is neither pre-trained nor updated online. To evaluate different algorithms more comprehensively, we additionally add a new Nash opponent. This opponent's policy is a part of an approximate Nash Equilibrium generated iteratively by the CFR (Zinkevich et al., 2008) algorithm. Playing a strategy from a Nash Equilibrium in a two-player zero-sum game is guaranteed not to lose in expectation even if the opponent is the best response strategy when the value of the game is zero. We show the performance of the various algorithms in Table 1. It is clear that L2E maintains the highest profitability against all four types of opponents other than the random type. L2E can exploit the opponent with evident style significantly, such as the the Call opponent. Compared to other baseline methods, L2E achieved the highest average return against opponents with unclear styles, such as the Rocks opponent, the Nash opponent, and the cheating Oracle opponent.

### 3.3 ABLATION STUDIES

#### 3.3.1 EFFECTS OF THE DIVERSITY-REGULARIZED POLICY OPTIMIZATION

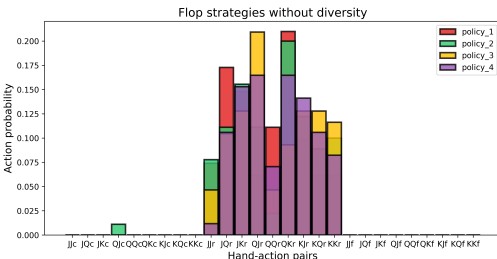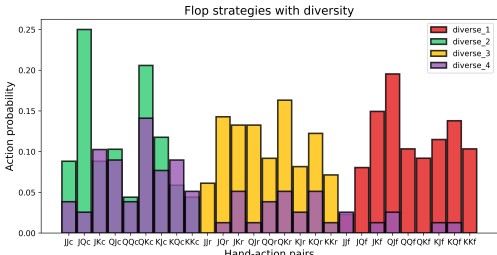

Figure 3: Visualization of the styles of the strategies generated with or without the MMD regularization term in the Leduc poker environment.

In this section, we verify whether our proposed diversity-regularized policy optimization algorithm can effectively generate policies with different styles. In Leduc poker, hand-action pairs represent different combinations of hands and actions. In the pre-flop phase, each player's hand has three possibilities, *i.e.*, J, Q, and K. Meanwhile, each player also has three optional actions, *i.e.*, Call (c), Rise (r), and Fold (f). For example, 'Jc' means to call when getting the jack. Action probability is the probability that a player will take a corresponding action with a particular hand. Fig. 3 demonstrates that without the MMD regularization term, the two sets of strategies generated in both the pre-flop and flop phases have similar styles. By optimizing with the MMD regularization term, the generated strategies are diverse enough which cover a wide range of different states and actions.

#### 3.3.2 EFFECTS OF THE HARD-OSG AND THE DIVERSE-OSG

As discussed previously, a crucial step in L2E is the automatic generation of training opponents. The Hard-OSG and Diverse-OSG modules are used to generate opponents that are difficult to exploit and diverse in styles. Fig. 4 shows the impact of each module on the performance of L2E. 'L2E w/o counter' is L2E without the Hard-OSG module. Similarly, 'L2E w/o diverse' is L2E without the Diverse-OSG module. 'L2E w/o diverse&counter' removes both modules altogether. The results show that both Hard-OSG and Diverse-OSG have a crucial influence on L2E's performance. It is clear that the Hard-OSG module helps to enhance the stability of the base policy, and the Diverse-OSG module can improve the base policy's performance significantly. To further demonstrate the generalization ability of L2E, we conducted a series of additional experiments on the BigLeduc poker and a Grid Soccer game environment (*c.f.* Appendix E and Appendix F for details).

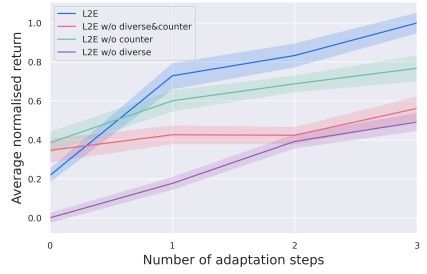

Figure 4: Each curve shows the average normalized returns of the base policy trained with different variants of L2E in the grid soccer environment.

## 4 CONCLUSION

We propose a learning to exploit (L2E) framework to exploit sub-optimal opponents without building explicit opponent models. L2E acquires the ability to exploit opponents by a few interactions with different opponents during training, so that it adapts to new opponents during testing quickly. We propose a novel opponent strategy generation algorithm that produces effective training opponents for L2E automatically. We first design an adversarial training procedure to generate challenging opponents to improve L2E's robustness effectively. We further exploit a diversity-regularized policy optimization procedure to generate diverse opponents to improve L2E's generalization ability significantly. Detailed experimental results in three challenging environments demonstrate the effectiveness of the proposed L2E framework.

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

## A   RELATED WORK

### A.1   OPPONENT MODELING

Opponent modeling is a long-standing research topic in artificial intelligence, and some of the earliest works go back to the early days of game theory research (Brown, 1951). The main goal of opponent modeling is to interact more effectively with other agents by building models to reason about their intentions, predicting their next moves or other properties (Albrecht & Stone, 2018). The commonly used opponent modeling methods can be roughly divided into four categories: policy reconstruction, classification, type-based reasoning and recursive reasoning. Policy reconstruction methods (Mealing & Shapiro, 2015) reconstruct the opponents' decision making process by building models which make explicit predictions about their actions. Classification methods (Weber & Mateas, 2009; Synnaeve & Bessiere, 2011) produce models which assign class labels (*e.g.*, "aggressive" or "defensive") to the opponent and employ a precomputed strategy which is effective against that particular class of opponent. Type-based reasoning methods (He et al., 2016; Albrecht & Stone, 2017) assume that the opponent has one of several known types and update the belief using the new observations obtained during the real-time interactions. Recursive reasoning based methods (Muise et al., 2015; de Weerd et al., 2017) model the nested beliefs (*e.g.*, "I believe that you believe that I believe...") and simulate the reasoning processes of the opponents to predict their actions. Different from these existing methods which usually require a large amount of interactive data to generate useful opponent models, our L2E framework does not explicitly model the opponent and acquires the ability to exploit different opponents by training with limited interactions with different styles of opponents.

### A.2 META-LEARNING

Meta-learning is a new trend of research in the machine learning community which tackles the problem of learning to learn (Hospedales et al., 2020). It leverages past experiences in the training phase to learn how to learn, acquiring the ability to generalize to new environments or new tasks. Recent progress in meta-learning has achieved impressive results ranging from classification and regression in supervised learning (Finn et al., 2017; Nichol et al., 2018) to new task adaption in reinforcement learning (Wang et al., 2016; Xu et al., 2018). Some recent works have also initially explored the application of meta-learning in opponent modeling. For example, the theory of mind network (ToMnet) (Rabinowitz et al., 2018) uses meta-learning to improve the predictions about the opponents' future behaviors. Another related work (Al-Shedivat et al., 2018) uses meta-learning to handle the non-stationarity problem in the multi-agent interactions. Different from these methods, we focus on how to improve the agents' ability to quickly adapt to different and unknown opponents.

### A.3 STRATEGY GENERATION

The automatic generation of effective opponent strategies for training with is a critical step in our approach, and how to generate diverse strategies has been preliminarily studied in the reinforcement learning community. In specific, diverse strategies can be obtained in a variety of ways, including adding some diversity regularization to the optimization objective (Abdullah et al., 2019), randomly searching in some diverse parameter space (Plappert et al., 2018; Fortunato et al., 2018), using information-based strategy proposal (Eysenbach et al., 2018; Gupta et al., 2018) and searching diverse strategies with evolutionary algorithms (Agapitos et al., 2008; Wang et al., 2019; Jaderberg et al., 2017; 2019). More recently, the researchers from DeepMind propose a league training paradigm to obtain a Grandmaster level StarCraft II AI (*i.e.*, AlphaStar) by training a diverse league of clorntinually adapting strategies and counter-strategies (Vinyals et al., 2019). Different from AlphaStar, our opponent strategy generation algorithm exploits adversarial training and diversity-regularized policy optimization to produce challenging and diverse opponents respectively.

## B ALGORITHM

---

**Algorithm 1:** The base policy training procedure of our L2E framework.

---

**Input:** Step size hyper parameters $\alpha, \beta$; base policy $B$ with parameters $\theta$; opponent policy $O$ with parameters $\phi$.

**Output:** An adaptive base policy B with parameters $\theta$

randomly initialize $\theta, \phi$ ;

initialize policy pool $\mathcal{M} = \{O\}$ ;

**for** $1 < e \leq epochs$ **do**

    $O = $ **Hard-OSG**$(B)$         $\triangleright$(see Alg. 2 ) ;

    $\mathcal{P} = $**Diverse-OSG**$(B, O, N)$     $\triangleright$(see Alg. 3 ) ;

    Update opponent policy pool $\mathcal{M} = \mathcal{M} \cup \mathcal{P}$ ;

    Sample batch of opponents $O_i \sim \mathcal{M}$ ;

    **for** *Each opponent $O_i$* **do**

        Construct a single-player MDP $M_B^{O_i}$ ;

        Sample trajectories $\tau$ using $B$ against fixed opponent $O_i$ ;

        Use Eqn. (1) to update the parameters of $B$ to obtain an adapted policy $B^{O_i}$ ;

        Resample trajectories $\tau'$ using $B^{O_i}$ against $O_i$;

    Update the parameters $\theta$ of $B$ according to Eqn. (4);

---

## C MAXIMUM MEAN DISCREPANCY

We use the Maximum Mean Difference (MMD) (Gretton et al., 2007) metric to measure the differences between the distributions of trajectories induced by different opponent strategies.

---

**Algorithm 2:** Hard-OSG, the hard-to-exploit training opponent generation algorithm.

---

**Input:** The latest base policy $B$ with parameters $\theta$.

**Output:** A hard-to-exploit opponent $\widehat{O}$ for $B$.

Randomly initialize $\widehat{O}$'s parameters $\widehat{\phi}$ ;

**for** $1 \leq i \leq epochs$ **do**

$\quad$ Construct a single-player MDP $M_B^{\widehat{O}}$ ;

$\quad$ Sample a small number of trajectories $\tau \sim M_B^{\widehat{O}}$ using $B$ against $\widehat{O}$ ;

$\quad$ Use Eqn. (5) to update the parameters of $B$ to obtain an adapted policy $B^{\widehat{O}}$ ;

$\quad$ Sample trajectories $\tau' \sim M_{\widehat{O}}^{B^{\widehat{O}}}$ using $\widehat{O}$ against $B^{\widehat{O}}$ ;

$\quad$ Update the parameters $\widehat{\phi}$ of $\widehat{O}$ according to Eqn. (7) ;

---

**Algorithm 3:** Diverse-OSG, the proposed diversity-regularized policy optimization algorithm to generate diverse training opponents.

---

**Input:** The latest base policy $B$, an exsiting opponent $O_1$, the total number of opponents that to
$\quad\quad$ be generated $N$.

**Output:** A set of diverse opponent $S = \{O_m\}_{m=1}^{N}$

Initialize the opponent set $S = \{O_1\}$ ;

**for** $i = 2$ *to* $N$ **do**

$\quad$ Randonly initialize an opponent $O_i$'s parameters $\phi_i$ ;

$\quad$ **for** $1 \leq t \leq steps$ **do**

$\quad\quad$ Calculate the objective function $\mathcal{L}^S(\phi_i)$ according to Eqn. (12);

$\quad\quad$ Calucute the gradient $\nabla_{\phi_i} \mathcal{L}^S(\phi_i)$ (*c.f.*, Appendix C for details) ;

$\quad\quad$ Use this gradient to update $\phi_i$ ;

$\quad$ Update the opponent set $S$, *i.e.*, $S = S \cup O_i$

---

**Definition 1** Let $\mathcal{F}$ be a function space $f : \mathcal{X} \rightarrow \mathbb{R}$. Suppose we have two distributions $p$ and $q$, $X := \{x_1, ..., x_m\} \sim p$, $Y := \{y_1, ..., y_n\} \sim q$. The MMD between $p$ and $q$ using test functions from the function space $\mathcal{F}$ is defined as follows:

$$\mathrm{MMD}[\mathcal{F}, p, q] := \sup_{f \in \mathcal{F}} \left( \mathbf{E}_{x \sim p}[f(x)] - \mathbf{E}_{y \sim q}[f(y)] \right). \tag{13}$$

If we can pick a suitable function space $\mathcal{F}$, we get the following important theorem (Gretton et al., 2007).

**Theorem 1** Let $\mathcal{F} = \{f \mid \|f\|_{\mathcal{H}} \leq 1\}$ be a unit ball in a Reproducing Kernel Hilbert Space (RKHS). Then $\mathrm{MMD}[\mathcal{F}, p, q] = 0$ if and only if $p = q$.

So the MMD distance between two strategies is 0 when the distributions of trajectories induced by them are identical. To obtain a set of strategies with diverse styles, we should increase the MMD distances between different strategies. $\varphi$ is a feature space mapping from $x$ to RKHS, we can easily

---

**Algorithm 4:** The testing procedure of our L2E framework.

---

**Input:** Step size hyper parameters $\alpha$; The trained base policy $B$ with parameters $\theta$; an
$\quad\quad$ unknown opponent $O$.

**Output:** The updated base policy $B^O$ that has been adapted to the opponent $O$.

Construct a single-player MDP $M_B^O$ ;

**for** $0 < step \leq steps$ **do**

$\quad$ Sample trajectories $\tau$ from the MDP $M_B^O$ ;

$\quad$ Update the parameters $\theta$ of $B$ according to Eqn. (1);

---

calculate the MMD distance using the kernel method $k(x, x') := \langle \varphi(x), \varphi(x') \rangle_{\mathcal{H}}$:

$$
\begin{aligned}
\text{MMD}^2 &(\mathcal{F}, p, q) \\
&= \|\mathbb{E}_{X \sim p} \varphi(X) - \mathbb{E}_{Y \sim q} \varphi(Y)\|_{\mathcal{H}}^2 \\
&= \langle \mathbb{E}_{X \sim p} \varphi(X) - \mathbb{E}_{Y \sim q} \varphi(Y), \mathbb{E}_{X \sim p} \varphi(X) - \mathbb{E}_{Y \sim q} \varphi(Y) \rangle \\
&= \mathbb{E}_{X, X' \sim p} k(X, X') - 2 \mathbb{E}_{X \sim p, Y \sim q} k(X, Y) + \mathbb{E}_{Y, Y' \sim q} k(Y, Y').
\end{aligned}
\tag{14}
$$

The expectation terms in Eqn. (14) can be approximated using samples:

$$
\text{MMD}^2[\mathcal{F}, X, Y] = \frac{1}{m(m-1)} \sum_{i \neq j}^{m} k(x_i, x_j)
$$

$$
+ \frac{1}{n(n-1)} \sum_{i \neq j}^{n} k(y_i, y_j) - \frac{2}{mn} \sum_{i,j=1}^{m,n} k(x_i, y_j).
\tag{15}
$$

The gradient of the MMD term with respect to the policy's parameter $\phi_j$ in our L2E framework can be calculated as follows:

$$
\begin{aligned}
\nabla_{\phi_j} \text{MMD}^2(\text{T}_i, \text{T}_j) &= \nabla_{\phi_j} \text{MMD}^2(\{\tau \sim M_{O_i}^B\}, \{\tau \sim M_{O_j}^B\}) \\
&= \mathbb{E}_{\tau, \tau' \sim M_{O_i}^B} [k(\tau, \tau') \nabla_{\phi_j} \log(p(\tau) p(\tau'))] \\
&\quad - 2 \mathbb{E}_{\tau \sim M_{O_i}^B, \tau' \sim M_{O_j}^B} [k(\tau, \tau') \nabla_{\phi_j} \log(p(\tau) p(\tau'))] \\
&\quad + \mathbb{E}_{\tau, \tau' \sim M_{O_j}^B} [k(\tau, \tau') \nabla_{\phi_j} \log(p(\tau) p(\tau'))],
\end{aligned}
\tag{16}
$$

where $p(\tau)$ is the probability of the trajectory. Since $\text{T}_i = \{\tau \sim M_{O_i}^B\}$, $O_i$ is the known opponent policy that has no dependence on $\phi_j$. The gradient with respect to the parameters $\phi_j$ in first term is 0. The gradient of the second and third terms can be easily calculated as follows:

$$
\nabla_{\phi_j} \log(p(\tau)) = \sum_{t=0}^{T} \nabla_{\phi_j} \log \pi_{\phi_j}(a_t | s_t).
\tag{17}
$$

## D  LEDUC POKER

The Leduc poker generally uses a deck of six cards that includes two suites, each with three ranks (Jack, Queen, and King of Spades, Jack, Queen, and King of Hearts). The game has a total of two rounds. Each player is dealt with a private card in the first round, with the opponent's deck information hidden. In the second round, another card is dealt with as a community card, and the information about this card is open to both players. If a player's private card is paired with the community card, that player wins the game; otherwise, the player with the highest private card wins the game. Both players bet one chip into the pot before the cards are dealt. Moreover, a betting round follows at the end of each dealing round. The betting wheel alternates between two players, where each player can choose between the following actions: call, check, raise, or fold. If a player chooses to call, that player will need to increase his bet until both players have the same number of chips. If one player raises, that player must first make up the chip difference and then place an additional bet. Check means that a player does not choose any action on the round, but can only check if both players have the same chips. If a player chooses to fold, the hand ends, and the other player wins the game. When all players have equal chips for the round, the game moves on to the next round. The final winner wins all the chips in the game.

Next, we introduce how to define the state vector. Position in a poker game is a critical piece of information that determines the order of action. We define the button (the pre-flop first-hand position), the action position (whose turn it is to take action), and the current game round as one dimension of the state, respectively. In Poker, the combination of a player's hole cards and board cards determines the game's outcome. We encode the hole cards and the board cards separately. The amount of chips is an essential consideration in a player's decision-making process. We encode this information into two dimensions of the state. The number of chips in the pot can reflect the action history of both players. The difference in bets between players in this round affects the choice of action (the game goes to the next round until both players have the same number of chips). In summary, the state vector has seven dimensions, *i.e.*, button, to_act, round, hole cards, board cards, chips_to_call, and pot.

# E BigLeduc Poker

We use a larger and more challenging BigLeduc poker environment to further verify the effectiveness of our L2E framework. The BigLeduc poker has the same rules as Leduc but uses a deck of 24 cards with 12 ranks. In addition to the larger state space, BigLeduc allows a maximum of 6 instead of 2 raises per round. As shown in Fig. 5, L2E still achieves fast adaptation to different opponents. In comparison with other baseline methods, L2E achieves the highest average return in Table 2.

|  | Random | Call | Raise | Oracle |
|---|---|---|---|---|
| L2E | **0.82±0.28** | **0.74±0.22** | **0.68±0.09** | **-1.02±0.24** |
| MAML | 0.77±0.30 | 0.17±0.08 | -2.02±0.99 | -1.21±0.30 |
| Random | -0.00±3.08 | -0.00±2.78 | -2.83±5.25 | -1.88±4.28 |
| TRPO | 0.19±0.09 | 0.10±0.16 | -2.22±0.71 | -1.42±0.47 |
| TRPO (pretrained) | 0.36±0.42 | 0.23±0.21 | -2.03±1.61 | -1.69±0.69 |
| EOM | 0.56±0.43 | 0.15±0.13 | -1.15±0.12 | -1.63±0.62 |

Table 2: The average return of each method when performing rapid adaptation against different opponents in the BigLeduc poker environment.

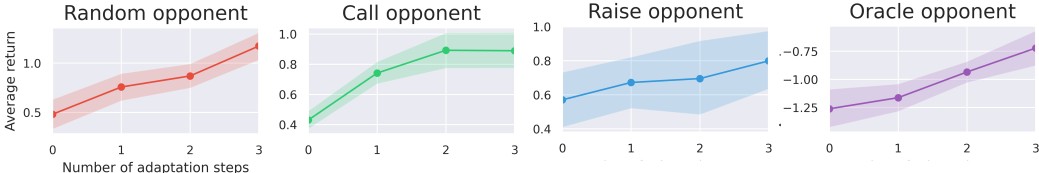

Figure 5: The trained base policy using our L2E framework can quickly adapt to different opponents of different styles and strengths in the BigLeduc poker environment.

# F Grid Soccer

This game contains a board with a $6 \times 9$ grid, two players, and their respective target areas. The position of the target area is fixed, and the two players appear randomly in their respective areas at the start of the game. One of the two players randomly has the ball. The goal of all players is to move the ball to the other player's target position. When the two players move to the same grid, the player with the ball loses the ball. Players gain one point for moving the ball to the opponent's area. The player can move in all four directions within the grid, and action is invalid when it moves to the boundary.

We train the L2E algorithm in this soccer environment in which both players are modeled by a neural network. Inputs to the network include information about the position of both players, the position of the ball, and the boundary. We provide two types of opponents to test the effectiveness of the resulting base policy. 1) A defensive opponent who adopts a strategy of not leaving the target area and preventing opposing players from attacking. 2) An aggressive opponent who adopts a strategy of continually stealing the ball and approaching the target area with the ball. Facing a defensive opponent won't lose points, but the agent must learn to carry the ball and avoid the opponent moving to the target area to score points. Against an aggressive opponent, the agent must learn to defend at the target area to avoid losing points. Fig. 6 shows the comparisons between L2E, MAML, and TRPO. L2E adapts quickly to both types of opponents; TRPO works well against defensive opponents but loses many points against aggressive opponents; MAML is unstable due to its reliance on task specification during the training process.

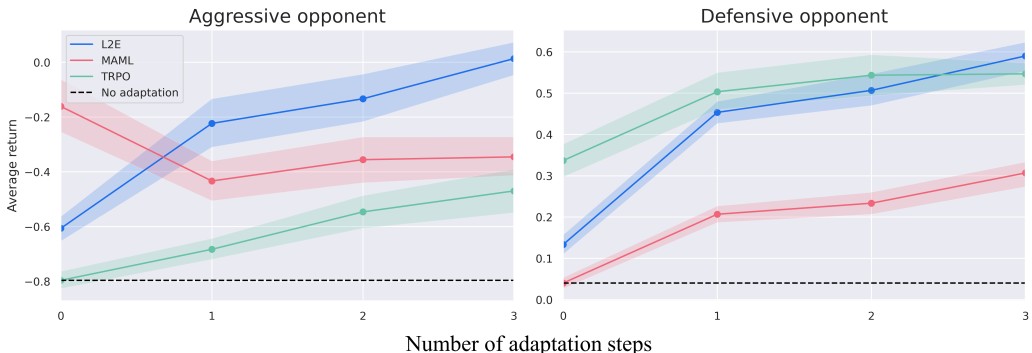

Figure 6: The trained base policy using our L2E framework can quickly adapt to opponents with different styles in a Grid Soccer environment.

## G    CONVERGENCE

Since convergence in game theory is difficult to analyze theoretically, we have designed a series of small-scale experiments to empirically verify the convergence of L2E with the help of Rock-Paper-Scissors(RPS) game. There are several reasons why RPS game is chosen: 1. RPS game is easy to analyze due to the small state and action space. 2. RPS game is easy to visualize and analyze due to the small state and action space. 3. RPS game is often used in game theory for theoretical analysis. The experiments we designed contains the following parts: 1. Testing the adaptability of Policy Gradient (PG), Self Play (SF), and L2E by visualizing the adaptation process. 2. Analyzing the relationship between L2E strategy and Nash strategy. 3. Analyzing the convergence of L2E.

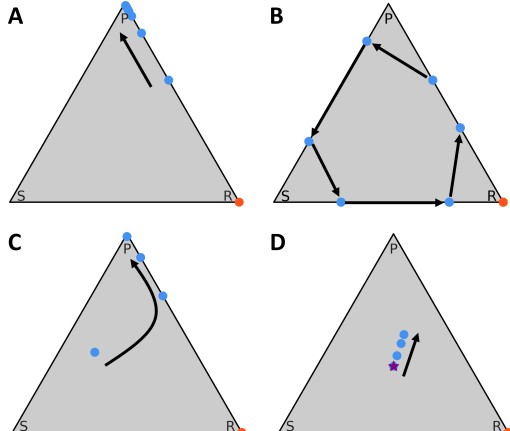

Figure 7: **A.** Policy Gradient : Optimize iteratively for the initial opponent (the orange dot), eventually converging to the best response of the initial opponent's strategy. **B.** Self Play : Each iteration seeks the best response to the **previous** round of strategy, which does not converge in an intransitive game like RPS. **C.** L2E's adaptation process when facing a new opponent (the orange dot). **D.** Nash policy's adaptation process when facing a new opponent (the orange dot).

As shown in Fig. 7, we can draw the following conclusions: 1. PG eventually converged to the best response, but it took dozens of gradient descent steps in our experiments (Each blue dot represents a ten-step gradient descent). SP failed to converge in the RPS game due to the intransitive nature of the RPS game (Rock>Scissors>Paper>Rock). In contrast, our L2E quickly converged to the best response strategy (Each blue dot represents a one-step gradient descent). 2. The strategy visualization in the Fig. 7.C shows that the base policy of L2E does not converge to the Nash equilibrium strategy after training but converges to the vicinity of the Nash equilibrium strategy. 3. If we fix

the base policy to the Nash strategy by imitation learning and then adapting it, we do not get good results either. This further illustrates the difference between the L2E strategy and the Nash equilibrium strategy. And the Fig. 8 further shows the performance of L2E and Nash strategy in RPS game when facing new opponents.

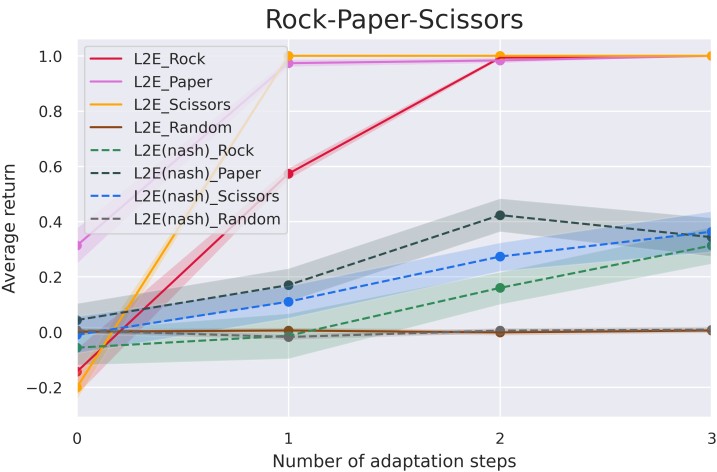

Figure 8: The performance of L2E and Nash strategy in RPS game when facing new opponents.

Although it is theoretically difficult to analyze the convergence properties of L2E, from the experimental results in Fig. 9, it can be seen that as the training progresses, L2E's adaptability becomes stronger and stronger. After reaching a certain number of iterations, the improvement eventually reaches a plateau, which provides some empirical evidence for the convergence of L2E.

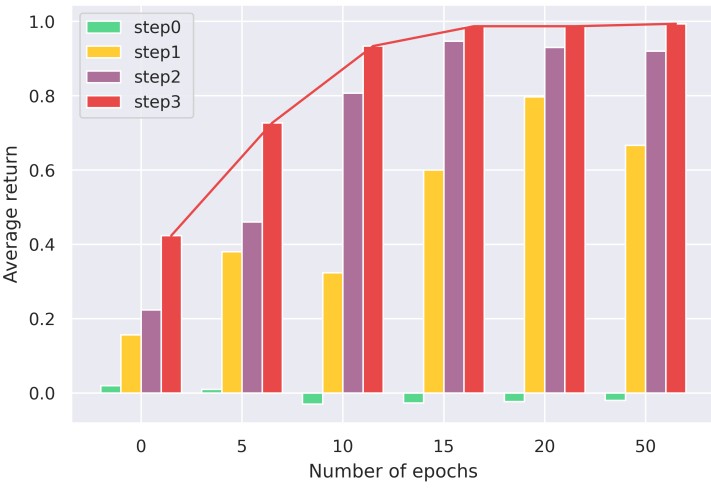

Figure 9: The convergence properties of L2E.

# H   HYPER-PARAMETERS

The hyper-parameters of all experiments are shown in the table below.

| Hyper-parameter | Value |
|---|---|
| Training step size hyper parameters $(\alpha, \beta)$ | (0.1,0.01) |
| Testing step size hyper parameter $\gamma$ | 0.1 |
| Number of opponents sampled per batch | 40 |
| Number of trajectories to sample for each opponent | 20 |
| Number of gradient steps in the training loop | 1 |
| Number of gradient steps in the testing loop | 3 |
| Policy network size(Leduc, BigLeduc, Grid Soccer) | [64,64,(4,4,5)] |
| Number of training steps required for convergence in Leduc | 300 |
| Number of training steps required for convergence in BigLeduc | 400 |
| Number of training steps required for convergence in GridSoccer | 300 |
| Hard-to-exploit opponent training epochs | 20 |
| Diverse opponent training epochs | 50 |
| Weight of the MMD term $\alpha_{\text{MMD}}$ | 0.8 |
| Bandwidth of RBF kernel | 1 |
| Minimum trajectory length N to calculate MMD term | 20 |
| Number of opponent strategies generated by OSG per round of iteration | N$\leq$5 |
| Number of trajectories sampled to compute MMD term | 8 |

Table 3: Hyper-parameters of L2E.

