# OpenReview forum: "L2E: Learning to Exploit Your Opponent"
_ICLR.cc/2021/Conference — Reject_

### Official Review · AnonReviewer2 · 2020-10-25
**Marginally above acceptance**

**Rating:** 6
**Confidence:** 3

**Review:**

Summary:
The authors propose an opponent modeling in 1-vs-1 games called the Learning to Exploit (L2E) framework, which exploits opponents by a few interactions with different opponents during training so that it can adapt to new opponents with unknown styles during testing quickly. In particular, the authors propose Opponent Strategy Generation (OSG) that produces effective opponents for training automatically through adversarial training for eliminating its own strategy’s weaknesses and diversity-regularized policy optimization to improve the generalization ability of L2E. Experimental results of two poker games and one grid soccer game indicate that L2E quickly adapts to diverse styles of unknown opponents.

Reasons for score:
Although motivation, solution, novelty, and overall presentation were almost clear, the implementation details were unclear and there were no shared codes. I think the idea contributed to this community, but mainly for the above reproducibility, it is difficult to provide a higher rating.

Pros:
1. Significance: the authors propose an opponent modeling called the L2E framework, which can adapt to unknown opponents quickly with a few observation of interactions
2. Methodological novelty: the authors propose OSG that produces effective opponents for training automatically through adversarial training and diversity-regularized policy optimization to improve the generalization ability of L2E.
3. Experiments: the results of two poker games and one grid soccer game indicate that L2E quickly adapts to diverse styles of unknown opponents.

Cons:
1. The implementation details were unclear (e.g., network and learning hyperparameters and their selections) and there were no shared codes.
2. There were some unclear points in Method and Experiments (below).

Other comments:

In Eqs. (3) and (4), the theta is updated based on (4) but also updated in computing pi_{theta O_i} described in the right-hand side of eq. (3). The former and the latter seem to update the theta in different manners, but the authors claimed that the theta is updated based on eq. (4). It seems to be confusing for me, but is this correct?

In Eqs. (7) and (8), notations of the loss and reward should be defined.

The experimental results were about the Leduc poker task, except for Fig. 4 (the Grid soccer task). There were no similar results of the Leduc poker task also in the Appendix. I would like to know the reason.

---

> ### Author Response · Authors · 2020-11-23
> **Response to AnonReviewer2**
>
> We sincerely appreciate your constructive and helpful comments. We initially address all your comments below:
>
> **Question 1** The Implementation details
>
> **Answer**: Thank you for your comments. We have added the hyper-parameters and training details of the experiments in the appendix.
>
> **Question 2**  The update manner of theta
>
> **Answer**: The optimization is performed over the base policy’s parameters $\theta$, whereas the objective function is computed using the adapted based policy’s parameters $\theta^{O_i}$. $\theta^{O_i}$ is also obtained from $\theta$ by a single gradient step. So, updating $\theta$ involves a gradient through a gradient. Computationally, this requires an additional backward pass to compute Hessian-vector products, which is supported by modern deep learning libraries such as TensorFlow and Pytorch.
>
> **Question 3** Notations of the loss and reward should be defined
>
> **Answer**:  Thank you for your instructive suggestion, we will add these two definitions in the revised manuscript.
>
> **Question 4** There were no similar results to the Leduc poker task in the Appendix
>
> **Answer**: The practice of showing experimental results in this way in our paper considers the different environmental characteristics of soccer and poker. The performance of the base policy and other baselines in the Grid Soccer environment against different opponents' styles can be clearly visualized in a single figure. This is because the rewards of the Grid Soccer environment are relatively deterministic, and a player scores one point for taking the ball to the opponent's goal. The average reward in a single game is limited to the range of [-1, 1]. In poker, however, the average rewards are not strictly limited to a suitable range due to the large uncertainty of the chips raised, so we use tables and figures to show the full effect of the poker experiment.

---

### Official Review · AnonReviewer4 · 2020-10-27
**Is L2E just finding a NE?**

**Rating:** 6
**Confidence:** 3

**Review:**

## Summary

The paper motivates the problem setting of quickly adapting to exploit sub-optimal opponents. The goal is to find a base policy which is quick to adapt to a range of sub-optimal opponents. To do this they use L2E which can generate exploiter and diverse opponent agents and updates the base policy based on trajectories from exploiter base policies. The paper focuses on the 2-player zero-sum setting where we have many algorithms which can find NE. I am concerned that this procedure is simply finding a NE and the other algorithms compared (MAML and TRPO) are not. I believe the paper needs to construct an argument that 1) L2E is not finding a NE and that 2) NE is not a good base policy.

=== Rebuttal Edit (Increased score form 4 to 6)

Thanks for the discussions. The primary reason for my score increase is discovering that the power of the framework is finding a representation that is quick at exploiting new opponents. I have been sufficiently convinced that this approach is not simply finding a NE (a sticking point in my review). I think that there are a collection of ideas here that are publishable and are of interest to the community.

The reason for not giving a higher score is that I think the points the paper made could be clearer: specifically I think the phrase "base policy" could be better replaced by "base representation" / "base model". I was stuck on the idea that the base policy had to be a strong one (eg a NE), and close to exploiter policies in *policy space* rather than *parameter space*. Re-reading after the paper update, I am worried that a significant portion of readers may fall into the same trap despite the authors' additional edits. Tightening the story would make this paper more appealing. I also broadly agree with the other reviewers suggestions / concerns.

For future work (also mentioned by another reviewer), I think there is no reason the "base policy" could not also be a strong policy too. I believe with minor adjustments to your framework this could be achieved, and one would have a model that both has low exploitability and is fast to adapt to new opponents - a potentially powerful combination.

## Score

I am recommending reject [UPDATE - see above] - although I will maintain an open mind due to my middling confidence in some of the background literature around the paper.

## Positives

Having a diversity regularized policy optimization procedure with MMD is interesting.

The PG update rules are interesting.

## Concerns

I am unsure about some of the claims wrt implicit/explicit opponent modelling. It seems that we need the exact opponent and base policy at all times in L2E. The literature that is cited in the paper is in a harder setting than this - where the opponent’s policy has to be estimated. Are we actually modelling opponents?

The MMD term seems very expensive and I am not sure it will scale, particularly with population size. It is also unclear to me how much sampling is necessary to estimate MMD. This approach does interest me however, perhaps the authors could comment on the scalability of this metric?

In the experiments section I would like to see more details on the actual training. How many outer and inner iterations were needed to converge? How many diverse and hard opponents were used during training? How many trajectories are sampled? Is learning rate tuning important? I am not sure I could reproduce from the details provided.

The results in Figure 2 do not seem to match up with the numbers in Table 1.

It is not clear whether the results are because the L2E procedure finds base policy close to Nash or if it is finding some other interesting policy. Would it be possible to adapt a Nash policy (with 4 adaptation steps) as an additional baseline for Table 1 and Figure 2 to check this? My concern is that the L2E framework is just finding Nash in an exotic way. From Table 1 it is clear the other baselines are not finding Nash. I appreciate that Nash was included as an opponent. I think testing this is key to back some of the claims made in this paper. If this procedure is indeed finding Nash, then is it doing it more efficiently than PSRO (Lanctot 2017) or self-play?

Zero-sum two-player settings NE is a reasonable thing to optimize for. Things like fictitious self play (FSP) are known to converge to approximate NE, and this algorithm looks like FSP without running the best response calculation to convergence.

Who is the opponent in Figure 4?

## Other Things

Figure 1: Should there be a loop from B’ back to B? Similar for O’ to O?

Page 2: “The key idea underlying L2E to train...” -> “L2E *is* to train”

Page 5: “only consider competitive agents in this work...” -> More specifically only zero-sum agents.

Page 7: “As with the previous experiments, we also use four gradient updates when adapting to a new opponent” -> The previous experiments used three updates?

Page 7: “Positive returns are also guaranteed against opponents without a clear style…” -> Are they really guaranteed? There are a lot of things like step size stochastic rollouts that make this statement tenuous.

---

> ### Author Response · Authors · 2020-11-23
> **Response to AnonReviewer4 (2/2)**
>
> **Question 5**  1. Is the base policy close to Nash? 2. Adapting a Nash policy (with 3 adaptation steps) as an additional baseline 3. Comparison of convergence with methods such as self play.
>
> **Answer**: We have designed a series of small-scale experiments to empirically verify the convergence of L2E with the help of Rock-Paper-Scissors (RPS) game.
>
> There are several reasons why RPS game is chosen:
>
> 1. The Nash equilibrium strategy profile of the RPS game is obvious and unique.
> 2. RPS game is easy to visualize and analyze due to the small state and action space.
> 3. RPS game is often used in game theory for theoretical analysis.
>
> The experiments we designed contains the following parts:
>
> 1. Testing the adaptability of Policy Gradient (PG), Self Play (SF), and L2E by visualizing the adaptation process.
> 2. Analyzing the relationship between L2E strategy and Nash strategy.
> 3. Analyzing the convergence of L2E.
>
> From the experimental results (https://i.imgur.com/Cm9SIny.png , you can click on this anonymous link or refer to Fig.7 in the Appendix), we can draw the following conclusions:
>
> 1. PG eventually converged to the best response, but it took dozens of gradient descent steps in our experiments (Each blue dot represents a ten-step gradient descent). SP failed to converge in the RPS game due to the intransitive nature of the RPS game (Rock>Scissors>Paper>Rock). In contrast, our L2E quickly converged to the best response strategy (Each blue dot represents a one-step gradient descent).
> 2. The strategy visualization in the figure shows that the base policy of L2E does not converge to the Nash equilibrium strategy after training but converges to the vicinity of the Nash equilibrium strategy.
> 3. If we fix the base policy to the Nash strategy by imitation learning and then adapting it, we do not get good results either. This further illustrates the difference between the L2E strategy and the Nash equilibrium strategy. And the Fig.8 in the Appendix (https://i.imgur.com/3RozieG.png) shows the performance of L2E and Nash strategy in RPS game when facing new opponents.
>
> **Question 6**  Who is the opponent in Figure 4?
>
> **Answer**: In Figure 4, we normalized the sum of the base policy's return when facing aggressive and defensive opponents in the soccer environment and mapped them to the range [0, 1]. Figure 6 shows the performance of the base policy when facing these two types of opponents, respectively.
>
> **Question 7** Figure 1: Should there be a loop from B’ back to B? Similar for O’ to O?
>
> **Answer**: Thank you for your suggestion, we will redraw Figure 1 in the revised manuscript.
>
> **Question 8** “As with the previous experiments, we also use four gradient updates when adapting to a new opponent” -> The previous experiments used three updates?
>
> **Answer**: Thank you for your comment. This is a mistake in our presentation. All adaptation processes in this paper are three-step gradient updates, and we have corrected this mistake.
>
> **Question 9** “Positive returns are also guaranteed against opponents without a clear style…” -> Are they really guaranteed? There are a lot of things like step size stochastic roll-outs that make this statement tenuous.
>
> **Answer**: Thank you for your valuable suggestion. We apologize for our inaccurate statement.  These are indeed empirical results but certainly not theoretical guarantees. We've used more precise expressions:  Compared to other baseline methods, L2E achieved the highest average return against opponents with unclear styles, such as the Rocks opponent, the Nash opponent, and the cheating Oracle opponent.

---

> > ### Comment · AnonReviewer4 · 2020-11-23
> > **Question 5**
> >
> > I think RPS is a good small game to explore this question (thank you for including this analysis in the rebuttal), however I am still confused by L2E's performance, and the claims. I hope that by quickly replying the authors will have time to clarify for me.
> >
> > Let me make some statements which I think are true (and please correct me if I am wrong) in order to help my understanding.
> >
> > 1. L2E is a procedure that a) finds a base policy b) can adapt to new agents quickly.
> > 2. For RPS the NE is [1/3, 1/3, 1/3].
> > 3. There are three obvious pure new/exploiter agents R, P, and S which are all easily exploitable with adaptation.
> >
> > Is seems to me intuitively (symmetry, has shortest max distance to R, P, S, etc...) that the most natural base policy in this game that could adapt to all possible new opponents quickly is the NE policy (as shown in Figure 7D as a purple star). Questions: 1) What is the base policy in Figure 7C?  2) Is it better at adapting individually to all three of R, P, and S than it would have if [1/3, 1/3, 1/3] was the base policy? If yes (I am very suspect), is there an intuitive explanation? 3) Are 7C and 7D using the exact same adaptation process to produce the blue points? 4) Why are steps in 7D much slower than 7C?
> >
> > To clarify in my review above I would (intuitively) expect L2E to have converged to [1/3, 1/3, 1/3] for the base policy in RPS. The thrust of my review was trying to work out if "find a base policy" step is just a fancy NE finder. In RPS I am surprised it is 1) not finding NE as base policy and 2) that a non-NE base policy could faster adapt to all R, P, S agents.

---

> > > ### Author Response · Authors · 2020-11-23
> > > **Response to AnonReviewer4 (The new question)**
> > >
> > > We sincerely appreciate your timely constructive and helpful comments.
> > >
> > > **Question 1:** What is the base policy in Figure 7C?
> > >
> > > **Answer:** The base policy in Figure 7C is obtained by our L2E framework. It is a neural network with special parameters which can fast adapt to different opponents. In contrast, the Nash base policy is a network with random parameters which outputs R, P and S with equal probability, since the Nash equilibrium of RPS is [1/3,1/3,1/3].
> > >
> > > **Question 2:** Is it better at adapting individually to all three of R, P, and S than it would have if [1/3, 1/3, 1/3] was the base policy? If yes, is there an intuitive explanation?
> > >
> > > **Answer:**  Yes.  Figure 8 in the Appendix (https://i.imgur.com/3RozieG.png) shows the results of L2E's base policy and the Nash base policy adapt individually to all three of R, P, and S. It is clear that L2E's base policy can adapt to different opponents more quickly than the Nash base policy. These seemingly abnormal results are actually easy to explain. The key idea underlying L2E is to train a base policy against various styles of opponents by using only **a few interactions** between them during training, such that it acquires the ability to exploit different opponents quickly. **After training, the base policy can quickly adapt to new opponents using only a few interactions during testing.** In other words, the base policy is **explicitly trained** such that a few interactions with a new opponent will produce an opponent-specific policy to effectively exploit this opponent. **L2E's learning process can be viewed as maximizing the sensitivity of the rewards obtained when facing new opponents with respect to the parameters: when the sensitivity is high, small local changes to the parameters can lead to large improvements in the obtained rewards.** Our L2E can be seen as a special case of learning to learn or meta-learning, i.e., **it learns to fast adapt to its opponents.** In contrast, the Nash base policy is essentially a random strategy in this example, therefore, this **vanilla random policy** without explicitly trained to fast adapt to its opponents does not have the ability to quickly adapt to its opponents. Of course the Nash base policy can finally converge to the best response strategy of its opponent, but it is much slower than L2E's base policy.  We hope this explanation can address your concerns.
> > >
> > > **Question 3:** Are 7C and 7D using the exact same adaptation process to produce the blue points?
> > >
> > > **Answer:** Yes, the experimental settings of 7C and 7D are exactly the same.
> > >
> > > **Question 4:** Why are steps in 7D much slower than 7C?
> > >
> > > **Answer:** Please refer to our answer to Question 2.

---

> > > > ### Comment · AnonReviewer4 · 2020-11-23
> > > > **Function Approximator is the Secret Sauce**
> > > >
> > > > Thank you for your quick response - this plugs a big gap in my understanding. The part I was missing was that fast adaptation is encoded in the function approximator which becomes parameterized in such a way that it can converge faster to R, S, P with some constant learning rate. The means that the action probabilities of the base policy is not as important as it first seems. If the policy were instead coded with 3 parameters, rather than a NN, I imagine the 7C and 7D would then look more similar.

---

> > > > > ### Author Response · Authors · 2020-11-23
> > > > > **Response to AnonReviewer4 (The new question)**
> > > > >
> > > > > We are glad that our explanations address your concerns! Thank you again for your valuable comments.

---

> > > > > ### Comment · AnonReviewer3 · 2020-11-25
> > > > > **I think this discussion is essential for follow-up work**
> > > > >
> > > > > Thanks to Reviewer 4 and authors for this discussion.
> > > > > I did not raise this point in my review because I think there were more "basic" and pressing points to be addressed for the paper in its current form.
> > > > >
> > > > > However, I had got to similar conclusions. This may not be true for more complex games where the landscape is not nearly as regular or symmetrical, but at least for RPS, I would actually be very surprised if it was not possible to learn an arbitrarily good approximation of the NE (which is obviously very different than the one the authors learnt by imitation learning, mostly because the network is very over-parameterized) that is such that it can adapt at least as fast as LE to arbitrary opponents.
> > > > > In that sense, I think that LE can be seen as some sort of regularization technique to learn an implict opponent model that can quickly adapt. And then, provided that such "regularity" can be obtained on a good approximation of an NE, I don't really see why the base policy should be anything else than a NE.
> > > > >
> > > > > Funnily, that actually matches what most  the best human poker players have been trying to do, mostly using "solver" tools, in the last years, at least in cash games. They are implicitly trying to "implement" some sort of abstraction of the game in which they both can have a decent estimate of the NE and rapidly detect and adapt to the deviation of their opponents to that (perceived) NE.

---

> > > > > > ### Author Response · Authors · 2020-11-25
> > > > > > **The relationship between L2E and Nash**
> > > > > >
> > > > > > Thanks to reviewers 3 and 4 for their questions about the relationship between L2E and Nash. This question makes us think more deeply about why L2E works. Our L2E acquires the ability to exploit opponents by a few interactions with different opponents during training so that it can adapt to new opponents with unknown styles during testing quickly. After training, the base policy is a neural network with special parameters which can fast adapt to different opponents. We totally agree with Reviewer 3 that L2E can be seen as some sort of regularization technique to learn an implicit opponent model that can quickly adapt. From the simple RPS game, the trained base policy is indeed not a Nash strategy, since its output is not always [1/3,1/3,1/3]. The poker example mentioned by Reviewer 3 is very interesting. Humans adopt the Nash strategy because it is a safe strategy when the opponent is unknown, and humans have the remarkable ability to quickly change their initial Nash strategies to new ones. Our L2E gives the base policy the same ability of fast adaptation, however, from the current experimental results, the trained base policy is not necessarily a Nash before adaption. Studying the relationship between L2E's base policy and Nash equilibrium is a very interesting and exciting future work. We thank the reviewers again for this insightful question!

---

> ### Author Response · Authors · 2020-11-23
> **Response to AnonReviewer4 (1/2)**
>
> We sincerely appreciate your constructive and helpful comments. We initially address all your comments below:
>
> **Question 1** About implicit/explicit opponent modeling
>
> **Answer**: Our L2E framework does not explicitly construct an opponent model in the same way as the explicit opponent modeling approaches. During training, the opponents automatically generated by OSG are only used to improve the base policy's adaptability. During testing, our L2E framework does not require prior knowledge of the opponent type. A fully trained base policy can adapt quickly with a few gradient step updates when facing an unknown opponent type. This has also been verified by the experimental results in our paper.
>
> **Question 2** About the scalability of MMD
>
> **Answer**: According to the theoretical guarantee of Theorem 1, it is practically tractable to compute the MMD term, especially using the kernel method to simplify the computation. In fact, we found that setting the number of opponent strategies generated per iteration to N<5 across all environments and the number of trajectories sampled to compute MMD to 8, yields promising results. Thus, the computational burden of using the MMD term to encourage the generation of diverse opponent strategies is relatively small. We have added the other details of the calculation of the MMD term in the appendix.
>
> **Question 3** More details on the actual training
>
> **Answer**  Thank you for your comments. We have added the hyper-parameters and training details of the experiments in the appendix.
>
> **Question 4** The results in Figure 2 do not seem to match up with the numbers in Table 1
>
> **Answer**: In fact, the results of L2E against Oracle-type opponents in Table 1 were incorrect, and Figure 2 shows the correct raw data. We have carefully checked and re-run the code, and it was indeed our oversight that caused this error. We have corrected it in the revised manuscript.

---

### Official Review · AnonReviewer3 · 2020-10-28
**A stimulating idea that misses the mark in its current form.**

**Rating:** 4
**Confidence:** 3

**Review:**

## Summary
The paper suggests a novel framework (coined L2E) to learn a policy that is optimized to adapt quickly (and exploit) a wide range of unknown opponents.
To do so it trains a base policy that is optimized so that it can maximize its expected reward against a variety of opponents using only a few updates of its parameters (i.e. a few gradient steps of a straightforward optimization problem).
The various opponents that are used for the training of this base policy are generated in two steps.
First, given a base policy, an "hard-to-exploit" opponent is generated adversarially (in a procedure coined hard-OSG), to minimize the the reward that the base policy would get by adapting to it (using the few updates that are allowed to it in its adaptation step).
Then, given a base policy and a "hard-to-exploit" opponent, more diverse opponents are sequentially generated (in a procedure coined diverse-OSG) by optimizing their expected reward against the base policy while maximizing their "diversity" with the "hard-to-exploit" opponent and the already generated diverse opponents.
More formally, this diversity between two policies is defined (and optimized) as the MMD between the distributions over the trajectories that they generate (when the policies are seen as MDPs "playing" against the given base policy).
The base policy is then trained iteratively (as described above) against the diverse opponents, that are themselves generated (as described above) with the current iterate of the base policy.
After exposing this training procedure, the authors evaluate L2E on 3 toy games, showing that the trained policies are indeed able to benefit from little adaptations to a variety of heuristic opponents and perform better than some baseline methods.
They also empirically confirm that their "diversity-regularized policy optimization" indeed generates diverse policies.
Last, the authors empirically show the effect of their hard-OSG and diverse-OSG modules on the performance of L2E.

## Pros
- This paper is tackling a very relevant, interesting and difficult problem.
- I find the general approach of optimizing a policy to be able to "rapidly" and exploit a broad range of opponents to be very exciting.
- To the best of my (admittedly limited) knowledge, the suggested approach is significantly novel.
- While maybe a bit "roughly used" the diversity-inducing regularization term, using the MMD over the distribution of trajectories induced by the policies is interesting and potentially has a broader applicability than only L2E.

## Cons
- After careful reading, several key points remain unclear to me. Most notably, after training of L2E and when facing opponents with unknown policies, how does the base policy adapts? Is it done using eq. 2? If yes, is the expectation over the trajectories approximated with the actual observations made during the observation? How many observations are being used? If my understanding is correct, clarifying those points would help put in perspective how fast it actually takes for L2E to adapt in practice.
- It looks (to me, because no comment is made about it in the manuscript) like L2E must scale terribly with the size of the action space. First it must be extremely computationally intensive. While this remains feasible for the toy games that were used in the experiments, I am having very high doubts that this would scale well with larger games (even BigLeduc poker is ridiculously tiny compared to actual poker). At least, some comments about the computational aspects of L2E, or empirical evidence that L2E can handle larger games would be nice.
- A point is made, several times in the manuscript, that the base policy becomes "more robust and eliminates its weaknesses by learning to exploit hard opponents". First, it is not really clear what is precisely meant by this. Without further assumptions on the class of games, I do not really see why the base policy would be having a good expected reward before adaptation (either in average over a broad class of opponents or against the optimal opponent), or even less why it would be hard to exploit (especially after adaptation). In fact the empirical results suggest that the base policy is breaking even against a random opponent (before adaptation) at Leduc poker, which seems rather weak to me.
- The last point brings to a more general issue. I understand that the value of the contribution is more empirical than theoretical. Yet, it is absolutely not obvious to me whether L2E is supposed to converge at all (let alone having a clear idea about to what kind of solution it would converge). I am not a specialist of game theory, but I understand this is likely a difficult setting to analyze. At least, empirical evidence on the convergence of L2E would, in my opinion, strengthen greatly the manuscript.
- I find that the writing could also be improved. While the algorithm is admittedly hard to fully describe in a very succinct way, the amount of repetions or redundancies in the first 6 pages suggests that L2E could be more concisely and sharply introduced. The split between the main text and the appendices seems a bit arbitrary to me as I definitely think more content about related work should be exposed in the main document. At the very least, appendices A and B should be referenced in the main text. As it stands, there is literally no indication in the paper that the related work section and the algorithms can be found in the supplementary materials. There are also a lot of imprecisions in the form of somewhat vague claims or missing important details. In addition to the ones already mentioned, I would for instance take the example of section 3.2.2. where it is not clear to me how the first policy of diverse-OSG is generated in the absence of the hard-OSG module. And in that same paragraph the bold statement that hard-OSG helps enhance the stability (what is meant by that exactly?) is not clear at all to me. It is also claimed in 3.2. that positive returns are guaranteed against opponent without a clear style (whatever that precisely means). I see rather mild empirical envidence of this but certainly not guarantees. Another minor point is that I find the Theorem 1 to be a bit weirdly formatted. I imagine the theorem is supposed to be the statement that MMD equals 0 iff the two distributions are equal, but then, the following sentence shuld be more clearly separated (as not part of the theorem) and it should be more clearly stated that the result is not a contribution by adding the reference where this result first appeared (Gretton et al. '07, I assume). If not, the derivation of the gradient computation does not really constitute a "theorem". Last, there are a number of typos, or verbs missing throughout the manuscript, that should be easy to fix (sorry, it's really not convenient for me to list them without line numbers...).
- I'm a bit puzzled by some implementation details of the diverse-OSG. Notably, there seems to be no weighing on the MMD term in eq. 11. That seems pretty arbitrary to me. Could you elaborate on that? More specifically, I would imagine that if the MMD ways too little, the generated policies will be roughly identical while they will be diverse but potentially arbitrarily bad (in terms of expected reward). Also, while I don't have an issue with the somewhat arbitrary choice of an RBF kernel, I am a bit more puzzled by the choice of a width of 1. But I could imagine I'm missing an argument as to why this is a good choice.
- In Table 1, L2E is reported to have a positive average return against the oracle, which is defined as "making decisions based on perfect information". It's not clearly described what those decisions are but unless they are pretty bad, there is no way L2E or any policy can win against it. (And it should be pretty easy to find and implement the optimal strategy for the perfect information game.)

## Reasons for score
While I really want to emphasize that the problem is very interesting and that I like the premise of L2E, I think the paper, in its current form, is missing the target.
The main reasons can already be found in the "cons" that I listed.
To elaborate a bit further, I think that either the selected games for the experiments are too small and toy-like for a purely empirical paper (in contrast with AlphaStar or Liberatus achieving superhuman performance at games like Starcraft 2 or heads-up no limit hold'em, although they definitely tackle a different, and probably simpler problem).
In this current form, I consider the experiments as a crude proof-of-concept, which could be totally fine if there were more theoretical analysis to support the suggested approach.

## Questions during rebuttal period
I think several questions have already been raised in the rest of my review.
Most importantly, I would really love to understand how the base policy is updated in a "real setting", after training (see my cons #1).

---

> ### Author Response · Authors · 2020-11-23
> **Response to AnonReviewer3 (2/2)**
>
> **Question 5-1**  Appendices A and B should be referenced in the main text.
>
> **Answer**: Thank you for your valuable suggestions, we will add references to A and B in the main text.
>
> **Question 5-2**  How the first policy of diverse-OSG is generated in the absence of the hard-OSG module.
>
> **Answer**: The first strategy is generated using the objective function in Eqn. (11) without the MMD term, since there is no known strategy initially and the MMD term cannot be calculated. Once the initial opponent policy is obtained, Eqn. (11) that incorporates the MMD term can be used to encourage the generation of diverse policies to the existing ones.
>
> **Question 5-3** Hard-OSG helps enhance the stability (what is meant by that exactly?)
>
> **Answer**: From the ablation experiments in Figure 4, the Hard-OSG module helps improve the performance of L2E. Specifically, the performance of 'L2E w/o diverse&counter' is unstable, e.g., its two-step adaptation performance is roughly the same as its one-step adaptation performance. 'L2E w/o counter' alleviated this problem, but its final performance is still weaker than L2E, and the improvement eventually reaches a plateau. With the addition of Hard-OSG, L2E achieves the best performance, and the improvement is faster and more stable.
>
> **Question 5-4** Positive returns are guaranteed against opponent without a clear style (whatever that precisely means)
>
> **Answer**: Thank you for your instructive suggestion. We apologize for our inaccurate statement.  Compared to other baseline methods, L2E achieved the highest average return against opponents with unclear styles, such as the Rocks opponent, the Nash opponent, and the cheating Oracle opponent. These are indeed empirical results but certainly not theoretical guarantees. We will revise this sentence in the new version.
>
> **Question 5-5** The Theorem 1 to be a bit weirdly formatted
>
> **Answer**: Thank you for your valuable suggestions, we have modified the format of Theorem 1 and added the references (Gretton et al. '07).
>
> **Question 5-6** There are a number of typos, or verbs missing throughout the manuscript
>
> **Answer**: Thank you for your valuable suggestions. We will carefully proofread the manuscript and fix all these typos.
>
> **Question 6-1** The weight of the MMD term
>
> **Answer**: Thank you for your constructive suggestions. In the current experiment, we have set the weight of the MMD term $\alpha_{mmd}$ = 0.8, and it performs very well across all environments. There is indeed a trade-off here regarding diversity and validity (in terms of the expected reward). We believe that better experimental results can be achieved if the MMD item's weights are carefully tuned for each environment.
>
> **Question 6-2** The RBF kernel and the kernel width
>
> **Answer**: The choice of optimal kernel and bandwidth is an active research problem (Gretton et al., 2012; Fukumizu et al., 2009). In practice, the kernel bandwidth is heavily domain-dependent. In our experiments, we found that simply setting the bandwidth to 1 produced satisfactory results.
>
> **Question 7** About the results in Table 1
>
> **Answer**: In fact, the results of L2E against Oracle-type opponents in Table 1 were incorrect, and Figure 2 shows the correct raw data. We have carefully checked and re-run the code, and it was indeed our oversight that caused this error. We have corrected it in the revised manuscript.
>
>
>
> Reference:
>
> Arthur Gretton, Karsten M Borgwardt, Malte J Rasch, Bernhard Sch¨olkopf, and Alexander Smola.
> A kernel two-sample test. The Journal of Machine Learning Research, 13(1):723–773, 2012.
>
> Kenji Fukumizu, Arthur Gretton, Gert R Lanckriet, Bernhard Sch¨olkopf, and Bharath K Sriperumbudur.
> Kernel choice and classifiability for rkhs embeddings of probability distributions. In
> Advances in neural information processing systems, pp. 1750–1758, 2009.

---

> > ### Comment · AnonReviewer3 · 2020-11-25
> > **Feedback on authors' response**
> >
> > Again, thank you for clarifying those points and for addressing some of my remarks in the form of changes in the updated manuscript.
> > I still believe the clarity can be improved by making the first half more concise, but I understand it is easier said than done given the short amount of time given for this rebuttal period.

---

> ### Author Response · Authors · 2020-11-23
> **Response to AnonReviewer3 (1/2)**
>
> We sincerely appreciate your constructive and helpful comments. We initially address all your comments below:
>
> **Question 1**  how does the base policy adapts? Is it done using Eqn. (2)?
>
> **Answer** Your understanding is correct. The key idea underlying L2E to train a base policy against various styles of opponents by using only **a few interactions** between them during training, such that it acquires the ability to exploit different opponents quickly. **After training, the base policy can quickly adapt to new opponents using only a few interactions during testing.** In other words, the base policy is explicitly trained such that a few interactions with a new opponent will produce an opponent-specific policy to effectively exploit this opponent. More specifically, when facing a new opponent $O_i$, the trained base policy $\pi_{\theta}$ is allowed to query a few trajectories $\tau$ to adapt to $O_i$. The adapted parameters $\theta^{O_i}$ of the base policy are computed using one or more gradient descent updates with the sample trajectories $\tau$ using Eqn. (2).  We have added the number of trajectories used for training and testing, as well as other details in the appendix. We have also added an algorithm in the Appendix on how to adapt to new testing opponents by using the trained base policy.
>
> **Question 2** The size of the action space.
>
> **Answer**: The most significant difference between opponents with different styles lies in the actions taken at different states. When the game's size is small, the OSG module can easily explore the state and action space to generate high quality and comprehensive opponent strategies. For large games, the performance of L2E can be maintained by optimizing the OSG module, e.g., by increasing the number of opponent strategies generated or appropriately exploiting human knowledge. Furthermore, the abstraction approach to reduce the state and action spaces has been successfully applied to solve huge and complex games (such as six-player no-limit Texas hold'em poker). This is what we will focus on in future work.
>
> **Question 3** The meaning of the base policy becomes "more robust and eliminates its weaknesses by learning to exploit hard opponents"
>
> **Answer**: We apologize for the misunderstanding caused by our unclear description. Our OSG algorithm generates challenging, i.e., hard-to-exploit opponents. This is similar to the *hard example mining* concept in the machine learning community. In machine learning, the hard examples represent the training data that are misclassified by the classifier. By training with these hard examples, the classifier will become more robust and accurate.  Analogously, the challenging opponents generated by our OSG are strategies that the current base policy can not exploit, and they represent the weakness of the base policy. The base policy's adaptability can be improved by training with these challenging opponents. In other words, the base policy becomes more robust, and its weaknesses will be eliminated.
>
> **Question 4** Whether L2E is supposed to converge at all
>
> You are absolutely right that convergence can be very difficult to analyze theoretically. We have designed a series of small-scale experiments to empirically verify the convergence of L2E with the help of Rock-Paper-Scissors (RPS) game.
>
> There are several reasons why RPS game is chosen:
>
> 1. RPS game is easy to analyze due to the small state and action space.
> 2. RPS game is often used in game theory for theoretical analysis.
>
> Although it is theoretically difficult to analyze the convergence properties of L2E, from the experimental results (https://i.imgur.com/HbaB31g.png, you can click on this anonymous link or refer to Fig.9 in the Appendix), it can be seen that as the training progresses, L2E's adaptability becomes stronger and stronger. After reaching a certain number of iterations, the improvement eventually reaches a plateau, which provides some empirical evidence for the convergence of L2E.

---

> > ### Comment · AnonReviewer3 · 2020-11-25
> > **Feedback on authors' response**
> >
> > First of all, thanks a lot for taking the time for clarifying those points.
> >
> > Question 1: So you use the same number of trajectories (20) to approximate the expectation in both training and testing, correct? I would imagine that this number should highly depend on the size of the game and a too low number could lead to very noisy estimates. It's also not obvious to me how to set the trade-off between number of gradient steps and number of samples in the gradient estimate. Did you experience, in practice, that the choice of that hyper-parameter had a big impact on the performance?
> >
> > Question 2: Fair enough. Any insight on the rate at which the number of strategies to sample w.r.t. the size of the game? Is the OSG a computational bottleneck in the overall procedure?
> >
> > Question 3: If my understanding is correct, the term weakness is here to be taken with a very specific meaning as it describes strategies against which the base policy cannot rapidly adapt to / exploit. So even if the base policy has no weakness (in that sense), it can still be almost arbitrarily bad (what I think would be the more usual meaning of weak) against an arbitrarily large set of strategies, before the adaptation steps, no? (At least, that is how I understand that LE may converge to solutions that are significantly different from an approximate NE.) Either way, I think it generates a bit of confusion because (at least for me).

---

> > > ### Author Response · Authors · 2020-11-25
> > > **Response to AnonReviewer3 (The new question)**
> > >
> > > Thank you very much for your valuable and constructive feedback. The issues you mentioned are very important and insightful, although the deadline is only one hour left, we will try our best to answer your questions.
> > >
> > > **Question 1:** So you use the same number of trajectories (20) to approximate the expectation in both training and testing, correct? Is this number highly depend on the size of the game? How to set the trade-off between number of gradient steps and number of samples in the gradient estimate? The choice of that hyper-parameter had a big impact on the performance?
> > >
> > > **Answer:** Yes, we used the same number of trajectories (20)  in both testing and training to approximate expectations since our L2E learns to fast adapt to its opponents with few interactions.
> > >
> > > We totally agree with you that the number of trajectories sampled is closely related to the size of the game, since the larger the game, the more trajectories are needed to explore the state and action spaces. In our current experiments, sampling 20 trajectories is sufficient to approximate the expectation for the medium or small scale environments in the paper. How to scale L2E for larger-scale games, such as no-limit Texas hold'em poker (some information abstraction method may be used to reduce the size of the game to a manageable size), is our next research focus.
> > >
> > > The trade-off between the number of gradient steps and the number of samples also depends on the size of the game. In small-scale games such as RPS, it is possible to reach the best response strategy with only one-step gradient update (see Figure 8). When the games become larger, more gradient steps are needed to obtain a good strategy. In summary, these parameters are closely related to the game size, and in our experiments, we have obtained satisfactory results using **the same set of parameters** in all the environments (Leduc, BigLeduc and Soccer).
> > >
> > > **Question 2:** Any insight on the rate at which the number of strategies to sample w.r.t. the size of the game? Is the OSG a computational bottleneck in the overall procedure?
> > >
> > > **Answer:** The number of strategies and trajectories to sample are actually highly empirical with no theoretical guidance. The set of parameters in our manuscript works well at the current scale. It is true that OSG is the most computational intensive part of the entire algorithm. However, in our current experiments, by limiting the number of opponents N generated per iteration and the number of iterations, OSG is not a computational burden and achieves good performance. It is undeniable that solving large scale games requires complex training algorithms and huge computational resources, such as CFR, PBT, and other algorithms. This problem can be alleviated by using techniques such as state or action space abstraction, etc. Continuing to optimize the performance of L2E for solving larger-scale problems is a very exciting direction, and we are currently working on it.
> > >
> > > **Question 3:** The term weakness
> > >
> > > **Answer:** Your understanding is absolutely correct. More accurately, the term "weakness" represents "the weakness of the base policy's adaptability", i.e., it describes strategies against which the base policy cannot rapidly adapt to / exploit. We are sorry for this confusion, and we will describe it more clearly in the revised manuscript.
> > >
> > > **Question 4:** The relationship between L2E and Nash
> > >
> > > **Answer:** Thanks to reviewers 3 and 4 for their questions about the relationship between L2E and Nash. This question makes us think more deeply about why L2E works. Our L2E acquires the ability to exploit opponents by a few interactions with different opponents during training so that it can adapt to new opponents with unknown styles during testing quickly. After training, the base policy is a neural network with special parameters which can fast adapt to different opponents. We totally agree with Reviewer 3 that L2E can be seen as some sort of regularization technique to learn an implicit opponent model that can quickly adapt. From the simple RPS game, the trained base policy is indeed not a Nash strategy, since its output is not always [1/3,1/3,1/3]. The poker example mentioned by Reviewer 3 is very interesting. Humans adopt the Nash strategy because it is a safe strategy when the opponent is unknown, and humans have the remarkable ability to quickly change their initial Nash strategies to new ones. Our L2E gives the base policy the same ability of fast adaptation, however, from the current experimental results, the trained base policy is not necessarily a Nash before adaption. Studying the relationship between L2E's base policy and Nash equilibrium is a very interesting and exciting future work. We thank the reviewers again for this insightful question!

---

### Official Review · AnonReviewer5 · 2020-11-02
**Interesting idea about opponent population generation but with concern on appropriate baseline comparisons**

**Rating:** 6
**Confidence:** 4

**Review:**

**Summary:**
This paper proposes the Learning to Exploit (L2E) framework that can quickly adapt to diverse opponent's unknown strategies. The main contributions of L2E include: 1. learning of the base model based on the optimization similar to MAML (Finn et al., ICML-17) to adapt to a new opponent after a few learning iterations (Section 2.1), 2. the generation of the hard-to-exploit opponent to robustly train the base model (Section 2.2), and 3. the generation of diverse opponent policies using the maximum mean discrepancy (MMD) metric (Section 2.3). Empirical results show that L2E can exploit diverse opponents in the Leduc poker, BigLeduc poker, and Grid Soccer domains.

**Reasons for Score:**
Overall, I vote for a score of 5. While the opponent strategy generation (OSG) algorithm with the counter adaptability and diverseness is an interesting idea, I am concerned about inappropriate baseline comparisons (please refer to Concerns and Questions below). After seeing the authors' responses to my concerns, I am open to raising my score.

**Pros:**
1. OSG removes the requirement of preparing the population or task distribution in meta-learning, which can be expensive.
2. Section 3.3.1 shows promising results that the proposed MMD regularization term can generate diverse opponents.

**Concerns and Questions:**
1. OSG, which generates a competitive and diverse opponent population, is a paper's main contribution. However, this paper compares the baselines trained based on the random opponent population, such as the MAML baseline in Section 3.2. Hence, it is unclear how much more effective and diverse opponents that OSG can generate compared to state-of-the-art opponent generation-based algorithms (e.g., population-based RL (Jaderberg et al., Science-19)).
2. An important claim of the paper is that explicit opponent modeling requires large sample complexity (Section 1). However, an explicit opponent modeling baseline is not compared in the experimental results. Pre-training an explicit opponent modeling method on the same opponent population generated by OSG and then comparing its adaptation against a new opponent will further highlight L2E's fast adaptation performance.
3. In MAML, a non-meta-learning baseline is compared by pre-training a policy on all of the meta-training tasks and then fine-tuning at a meta-test task (i.e., the "pretrained" baseline in MAML). However, the TRPO baseline in this paper does not perform the pre-training ("The TRPO baseline does not perform pre-training ..." in Section 3.2). Because L2E's base model is pre-trained using opponents generated by OSG, a more fair comparison is to pre-train the TRPO (possibly based on the random opponents) and then fine-tune against a new opponent.
4. In Figure 2, the trained base policy receives the negative return playing against the oracle opponent (possibly expected as the oracle opponent takes actions based on perfect information). But, why does L2E receive the positive return in Table 1?

**Additional Feedback:**
1. In the appendix, hyper-parameters/training details for L2E and the baselines (e.g., the number of trajectories for each adaptation, details on how the meta-training task distribution is constructed for MAML) are missing.
2. In Tables 1 and 2, adding the variance will be helpful. Specifying the number of random seeds used in the experimental results will also be helpful.
3. Algorithm 1-3 are for learning the base policy (i.e., the meta-training procedure). Adding an algorithm in the Appendix when adapting to a new opponent after the base policy training (i.e., the meta-testing procedure) will be helpful.
4. Adding an explanation about why the particular metric of MMD (instead of other metrics) is chosen to compare the difference between distributions will be helpful.
5. In Section 2.2.2, it is noted that "For trajectories with different length, we clip the long trajectory to the same length as the short one" for the MMD calculation. I wonder whether applying the masking based on the done signal from the environment can be better than applying the clipping.

**Reference:**

Chelsea Finn, Pieter Abbeel, Sergey Levine. Model-Agnostic Meta-Learning for Fast Adaptation of Deep Networks. ICML, 2017.

Max Jaderberg, Wojciech M. Czarnecki, Iain Dunning, Luke Marris, Guy Lever, Antonio Garcia Castaneda, Charles Beattie, Neil C. Rabinowitz, Ari S. Morcos, Avraham Ruderman, Nicolas Sonnerat, Tim Green, Louise Deason, Joel Z. Leibo, David Silver, Demis Hassabis, Koray Kavukcuoglu, Thore Graepel. Human-level performance in 3D multiplayer games with population-based reinforcement learning. Science, 2019.

**After rebuttal:**
The responses address most of my main concerns, and I have increased the rating from 5 to 6. As discussed during the rebuttal, in the future, having additional experiments that compare between OSG and other appropriate population generation baselines would be helpful.

---

> ### Author Response · Authors · 2020-11-23
> **Response to AnonReviewer5 (1/2)**
>
> We sincerely appreciate your constructive and helpful comments. We initially address all your comments below:
>
> **Question 1:**  It is unclear how much more effective and diverse opponents that OSG can generate compared to state-of-the-art opponent generation-based algorithms (e.g., population-based RL (Jaderberg et al., Science-19)).
>
> **Answer:**  The population-based RL proposed in (Jaderberg et al., Science-19) is based on the idea of  Population-Based Training (PBT) (Jaderberg et al., arXiv-17). PBT is not an opponent generation algorithm, and it is essentially a general *hyper-parameter optimization* method by training a large population of models. It uses an *exploit* operation to replace the current weights with those who have the highest performance in the rest of the population and uses an *explore* operation to perturb the hyper-parameters with noise randomly. The population in PBT is only for model selection and hyper-parameter optimization, and only the best-performing model is retained in the end.
>
> Different from PBT, the purpose of our OSG is to generate a large number of *training opponents* automatically. We can regard these training opponents as the *training data* of the base policy. The opponents generated by OSG are both sufficiently challenging (i.e., difficult to exploit) and sufficiently diverse. Challenging opponents are optimized in terms of the base policy through adversarial training, while diverse opponents are optimized in terms of the existing opponents through diversity-regularized policy optimization.
>
> In summary, the goals of our OSG and PBT are entirely different. In fact, PBT can be used as an automatic hyper-parameter tuning tool to further improve our algorithm's performance, which is an interesting future work.
>
> **Question 2:**  An explicit opponent modeling baseline is not compared in the experimental results.
>
> **Answer:** Thank you for your constructive comments. We have added a commonly used policy reconstruction based explicit opponent modeling baseline for comparison. This policy reconstruction based method explicitly fit an opponent model to reflect the opponent’s observed behaviors. In our implementation, we first collect interaction data for the same number of trajectories as other methods during the adaptation process. This interaction data is used to fit an opponent model.  The best response is then trained against the fitted opponent model. This best response is then used to interact with the opponent again to evaluate its performance. Even though we use more gradient steps to train the best response, the experimental results show that this explicit opponent modeling approach still performs worse than our L2E framework. The main problem with this approach is that relying on such a small amount of interaction data is simply not enough to train an accurate opponent model, resulting in an inaccurate best response. Depending on this inaccurate best response to make decisions does not work well.
>
> **Question 3:** Pre-train the TRPO (possibly based on the random opponents) and then fine-tune against a new opponent.
>
> **Answer:** Thank you for your instructive suggestion. we have updated the experimental results. It is indeed more fair to pre-train TRPO before comparing it to L2E.
>
> **Question 4**  The results in Figure 2 do not match up with the numbers in Table 1.
>
> **Answer:** In fact, the results of L2E against Oracle-type opponents in Table 1 were incorrect, and Figure 2 shows the correct raw data. We have carefully checked and re-run the code, and it was indeed our oversight that caused this error. We have corrected it in the revised manuscript.

---

> > ### Comment · AnonReviewer5 · 2020-11-24
> > **Feedback on Revision 1**
> >
> > I would like to thank the authors for carefully considering my initial concerns and making appropriate changes in the revision.
> > Most of my concerns are addressed, except Question 1:
> >
> > **Question 1:** It is unclear how much more effective and diverse opponents that OSG can generate compared to state-of-the-art opponent generation-based algorithms (e.g., population-based RL (Jaderberg et al., Science-19)).
> >
> > I apologize for the lack of clarity. I agree with the authors that [Population-Based Training (PBT) (Jaderberg et al., arXiv-17)](https://arxiv.org/pdf/1711.09846.pdf) is an automatic hyper-parameter optimization technique. However, in [the population-based RL paper (Jaderberg et al., Science-19)](https://arxiv.org/pdf/1807.01281.pdf), I was referring to their method in generating a population with diverse skills based on the concurrent training and Elo score computation (please refer to page 4). Regardless of whether the method in Jaderberg et al., Science-19 is a relevant baseline or not, I believe that it would be a valuable discussion to have whether OSG can generate a more effective/diverse population compared to other appropriate population generation baselines, which can further highlight the benefit of the proposed approach. I understand that it is not possible for the authors to run additional experiments given the rebuttal period. However, I hope the authors may consider my comment in the future.

---

> > > ### Author Response · Authors · 2020-11-25
> > > **Response to AnonReviewer5 (About question 1)**
> > >
> > > Thank you very much for your further elaboration and constructive feedback. First of all, we are glad that our responses address most of your concerns! After re-reading your suggested references, we agree that the PBT method can generate a population with diverse skills based on concurrent training and Elo score computation. In fact, we have already discussed different strategy generation methods in Appendix A.3, i.e., randomly searching a particular parameter space, using information theory-based strategy proposals, applying evolutionary algorithms, and so on. The PBT method can be classified as an evolutionary algorithm (please refer to page 4 in Jaderberg et al., arXiv-17, page 5 in Jaderberg et al., Science-19), and we have revised our manuscript to include references to the PBT method. Although both PBT and our OSG can generate a population of different policies, there are also obvious differences between them. PBT is to stabilize the learning process in partially observable multi-agent environments by concurrently training a diverse population of agents who learn by playing with each other, its ultimate goal is to find the optimal strategy for a specific task. While our OSG is specifically designed to generate effective training opponents to improve the base policy's adaptability. More specifically, the opponents generate by OSG are not only diverse but also challenging (hard-to-exploit). These diverse and hard-to-exploit training opponents are particularly suitable for improving the base policy's adaptability. Although there are obvious differences between PBT and our OSG in terms of goals and implementation details, PBT has achieved remarkable results on various complex tasks. Following your valuable suggestions, we promise to add some additional experiments to compare with PBT or some other appropriate population generation baselines in the final version.

---

> ### Author Response · Authors · 2020-11-23
> **Response to AnonReviewer5 (2/2)**
>
> **Question 5**   Hyper-parameters/training details for L2E and the baselines.
>
> **Answer:** Thank you for your comments. We have added the hyper-parameters and training details of all the algorithms in the appendix.
>
> **Question 6**  Adding the variance in Table 1 and 2, and specifying the number of random seeds used in the experimental results.
>
> **Answer:** Thank you for your valuable suggestion. We have added the variance to Tables 1 and 2. Each configuration is tested with five random seeds, and we have added the details of the other hyper-parameters in the appendix.
>
> **Question 7** The meta-testing procedure
>
> **Answer:** Thank you for your valuable suggestion. We will add the pseudo-code of the meta-testing procedure in the Appendix.  When facing a new opponent $O_i$, base policy $\pi_{\theta}$ is allowed to query a limited number of sample trajectories $\tau$ to adapt to $O_i$. The adapted parameters $\theta^{O_i}$ of the base policy are computed using one or more gradient descent updates with the sample trajectories $\tau$ using Eqn. (2). Our experimental results show that base policy can be very effective against new opponents by merely using a small number of steps in the adaptation process. We have added an algorithm in the Appendix on how to adapt to new opponents after base policy training.
>
>  **Question 8**  Why the particular metric of MMD is chosen
>
> **Answer:** We have already analyzed why we chose the MMD metric in section 2.2.2.  The MMD metric can better describe the differences between strategies than methods that focus only on state space or action space. Following your valuable suggestion, we have further elaborated on the MMD metric's advantages over other metrics. In contrast to the Wasserstein distance and Dudley metrics, the MMD metric has closed-form solutions. And unlike KL-divergence, the MMD metric is strongly consistent while exhibiting good rates of convergence (Sriperumbudur et al., 2010).
>
> **Question 9**  Masking or clipping
>
> **Answer:** We are sorry that the description in the original manuscript is misleading. In fact, we have set a minimum trajectory length $N$, and we only clip the trajectories when both of them are longer than $N$. Usually, the trajectory's length does not exceed N, and we apply the masking based on the done signal from the environment to make them the same length.
>
>
>
> Reference:
>
> Max Jaderberg, Valentin Dalibard, Simon Osindero, Wojciech M Czarnecki, Jeff Donahue, Ali
> Razavi, Oriol Vinyals, Tim Green, Iain Dunning, Karen Simonyan, et al. Population based training
> of neural networks. arXiv preprint arXiv:1711.09846, 2017.
>
> Max Jaderberg, Wojciech M. Czarnecki, Iain Dunning, Luke Marris, Guy Lever, Antonio Garcia Castaneda, Charles Beattie, Neil C. Rabinowitz, Ari S. Morcos, Avraham Ruderman, Nicolas Sonnerat, Tim Green, Louise Deason, Joel Z. Leibo, David Silver, Demis Hassabis, Koray Kavukcuoglu, Thore Graepel. Human-level performance in 3D multiplayer games with population-based reinforcement learning. Science, 2019.
>
> Bharath K Sriperumbudur, Kenji Fukumizu, Arthur Gretton, Bernhard Sch¨olkopf, and Gert RG
> Lanckriet. Non-parametric estimation of integral probability metrics. In 2010 IEEE International
> Symposium on Information Theory, pp. 1428–1432. IEEE, 2010.

---

### Decision · Program_Chairs · 2021-01-07
**Final Decision**

**Decision:**

Reject

**Comment:**

Well, this paper has achieved something remarkable in this review process:  The initial scores came in at fairly low scores (4, 5, 3, 6).  However, as the discussions / rebuttals went back and forth, the reviewers were able understand and see the merits of the proposed methodology.  Namely, the setting of L2E (Learning to Exploit), which makes use of a novel method called Opponent Strategy Generation, to quickly generate very different types of opponents to play against.  One more pertinent component is the use of MMD (maximum mean discrepancy regularization) which can remove the necessity of dealing with task distributions, and does a better job in creating diverse opponents.

Having understood the technical approach, three of the reviewers decided to substantially increase their scores. R4 increased 4->6, R5 increased 5->6, R3 increased 3->4, while R2 held steady with a score of 6. It was also good to see empirical favorable results compared to other baseline methods: L2E had the best return against unclear opponents, such as Rocks opponent and Nash opponent.

Without any reviewer arguing strongly for acceptance, the program committee decided that the paper in its current form does not quite meet the bar, and also that it would benefit from another revision.